# NAD-dependent dehydrogenases enable efficient growth of *Paracoccus denitrificans* on the PET monomer ethylene glycol

Minrui Ren[1], Danni Li ®[1], Holly Addison ®[2], Willem E. M. Noteborn ®[3], Elisabeth H. Andeweg[4], Timo Glatter ®[5], Johannes H. de Winde ®[1], Johannes G. Rebelein ®[2,6], Meindert H. Lamers ®[4] & Lennart Schada von Borzyskowski ®[1] ✉

Ethylene glycol is a monomer of the plastic polyethylene terephthalate (PET) and an environmental pollutant of increasing concern. Although it is generally accepted that bacteria use ethylene glycol as growth substrate, not all involved enzymes are well understood. Here, we show that *Paracoccus denitrificans* assimilates ethylene glycol solely via NAD-dependent alcohol and aldehyde dehydrogenases. Using comparative proteomics, we identify a gene cluster that is strongly expressed in the presence of ethylene glycol. We report the functional and structural characterization of EtgB and EtgA, key enzymes encoded by this *etg* gene cluster. We furthermore show that the transcriptional activator EtgR controls expression of the gene cluster. Adaptive laboratory evolution on ethylene glycol results in faster growth, enabled by increased production of EtgB and EtgA. Bioinformatic analysis reveals that the *etg* gene cluster is widely distributed among bacteria, suggesting a common role of NAD-dependent dehydrogenases in microbial ethylene glycol assimilation.

Polyethylene terephthalate (PET) is one of the most commonly used plastics, utilized in synthetic fibers, water containers, and food packaging. From the 1990s onward, the demand for PET, and therefore its production, increased exponentially. This increased usage of PET has resulted in a staggering accumulation of undegraded plastic waste. Nearly 80% of the 6.3 billion tons of plastic waste that had been generated as of 2015 were accumulated in landfills or the natural environment[1]. Moreover, the production of PET relies heavily on non-renewable fossil fuels, exacerbating environmental concerns over its widespread use[2]. In alignment with principles of environmental sustainability, the biotechnological upcycling of PET has recently emerged as a compelling solution[3]. Since the discovery of PETase, a hydrolase capable of depolymerizing this polyester[4], enzymatic plastic breakdown is increasingly considered as a promising solution for managing PET waste. This enzyme and its improved derivatives, such as FAST-PETase[5], Combi-PETase[6], or HotPETase[7], enable the breakdown of PET into bis(2-hydroxyethyl) terephthalate (BHET) and mono(2-hydroxyethyl) terephthalate (MHET). Subsequently, the enzyme MHETase is responsible for further degradation of MHET into ethylene glycol and terephthalic acid[4]. The metabolic capability of microorganisms to utilize these monomers of PET for growth has been explored in various biotechnological applications, especially in the context of bioremediation and bioconversion processes aimed at transforming plastic waste into useful products using genetically engineered bacteria[8–14].

Ethylene glycol is a two-carbon alcohol with two hydroxy groups (Fig. 1). Besides being a PET monomer, ethylene glycol is also used in other polyester resins and fibers[15] as well as antifreeze agents and

[1]Institute of Biology Leiden, Leiden University, Leiden, The Netherlands. [2]Max Planck Institute for Terrestrial Microbiology, Marburg, Germany. [3]Netherlands Center for Electron Nanoscopy, Leiden University, Leiden, The Netherlands. [4]Department of Cell and Chemical Biology, Leiden University Medical Center, Leiden, The Netherlands. [5]Facility for Mass Spectrometry and Proteomics, Max Planck Institute for Terrestrial Microbiology, Marburg, Germany. [6]Center for Synthetic Microbiology (SYNMIKRO), Philipps University Marburg, Marburg, Germany. ✉e-mail: L.Schada.von.Borzyskowski@biology.leidenuniv.nl

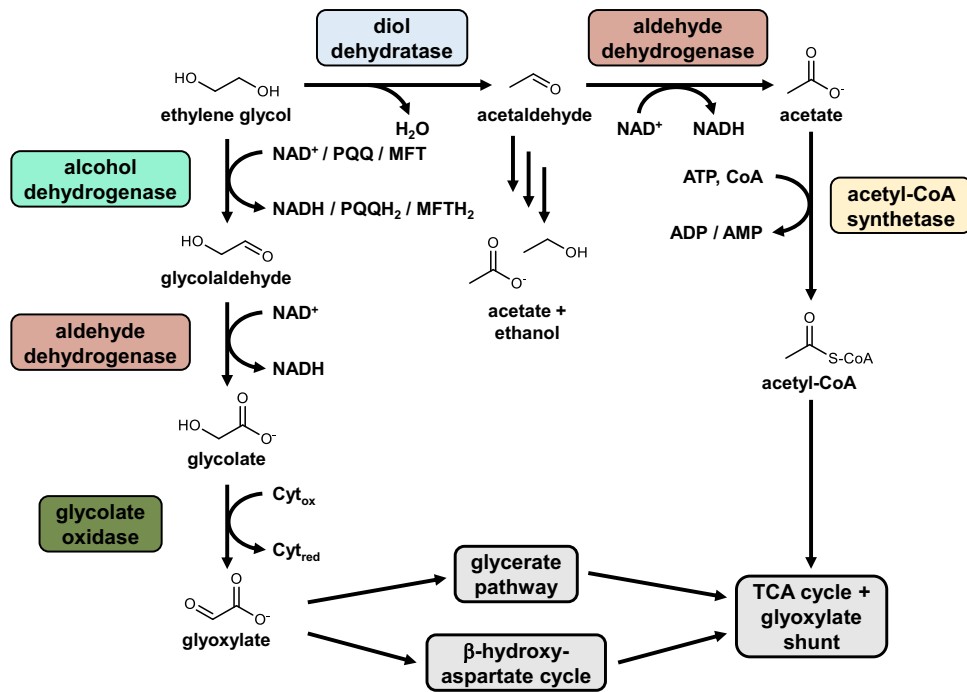

**Fig. 1 | Metabolic pathways involved in ethylene glycol assimilation.** Many bacteria oxidize ethylene glycol to glycolaldehyde using alcohol dehydrogenases with NAD[+], PQQ, or MFT as cofactor. Subsequently, glycolaldehyde is oxidized to glycolate and further to glyoxylate, which can enter central carbon metabolism via the glycerate pathway or the β-hydroxyaspartate cycle. In an alternative pathway, ethylene glycol is dehydrated to acetaldehyde, which is disproportionated into acetate and ethanol or converted into acetyl-CoA.

solvents. In 2022, the annual production capacity of ethylene glycol amounted to 57 million tons. Recently, it has gained increased attention as a next-generation feedstock in microbial biotechnology[16,17]. This compound is assimilated aerobically or anaerobically by microorganisms as a source of carbon and energy[18–21]. *Mycobacterium* sp. converts ethylene glycol to acetaldehyde via a coenzyme B12-dependent diol dehydratase, followed by further conversion to acetate[22]. The acetogen *Acetobacterium woodii* also dehydrates ethylene glycol to acetaldehyde, which is subsequently disproportionated to ethanol and acetate[23]. But in the most common metabolic pathway for ethylene glycol assimilation, the alcohol is initially oxidized to glycolaldehyde by alcohol dehydrogenases[18] (Fig. 1). In *Pseudomonas putida* JM37 and KT2440, the two pyrroloquinoline quinone (PQQ)-dependent enzymes PedE and PedH act as catalysts for this process[24–26]. In *Rhodococcus jostii* RHA1, mycofactocin (MFT) is the cofactor for the oxidation of ethylene glycol by an alcohol dehydrogenase[27,28]. However, PQQ- or MFT-dependent enzymes are limited to bacteria that possess the ability to synthesize the relevant cofactors; a large majority of bacteria only encode for nicotinamide adenine dinucleotide (NAD)-dependent alcohol dehydrogenases in their genomes. It is currently unclear to what extent NAD-dependent alcohol dehydrogenases enable efficient growth on ethylene glycol, since many characterized enzymes only exhibit poor kinetic parameters for the oxidation of this diol to glycolaldehyde (Supplementary Table 1). For example, the model bacterium *Escherichia coli* does not naturally grow on ethylene glycol; only an evolved strain with increased activity of the NADH-dependent lactaldehyde reductase FucO was capable of relatively slow growth (growth rate μ: 0.1 h[−1])[29,30].

Glycolaldehyde is further oxidized to glycolate by NAD-dependent aldehyde dehydrogenases. Glycolate is converted into glyoxylate by glycolate oxidase[18]. Finally, glyoxylate enters central carbon metabolism via the glyoxylate shunt[31] or dedicated pathways for net assimilation of glyoxylate (Fig. 1). To date, two net assimilation pathways for glyoxylate have been identified. The glycerate pathway is present in model bacteria, including *E. coli* and *P. putida*, and converts two molecules of glyoxylate

into one molecule of 2-phosphoglycerate[26,32,33]. Notably, the reaction sequence of this metabolic route involves the release of carbon dioxide and is energetically inefficient. The second pathway for net glyoxylate assimilation was first discovered in the 1960s in the Alphaproteobacterium *Paracoccus denitrificans*[34,35]; however, the comprehensive characterization of the β-hydroxyaspartate cycle (BHAC) was only completed much later[36]. This cyclic pathway catalyzes the energetically efficient conversion of two molecules of glyoxylate into the four-carbon compound oxaloacetate without the release of carbon dioxide. Due to its favorable characteristics, the BHAC has been introduced into the model plant *Arabidopsis thaliana* as an alternative photorespiration pathway[37] and into *P. putida* KT2440 as an alternative pathway for ethylene glycol assimilation. The *P. putida* strain engineered with the BHAC reached a growth rate of 0.24 h[−1], which was further increased to 0.31 h[−1] by adaptive laboratory evolution[12]. However, while the growth of *P. denitrificans* itself on ethylene glycol was recently demonstrated[38], the involved catabolic enzymes have not been characterized experimentally.

In this work, we use comparative proteome analysis to identify a gene cluster in *P. denitrificans* that includes genes encoding for a transcriptional regulator (which we termed EtgR), an alcohol dehydrogenase (which we termed EtgB), and an aldehyde dehydrogenase (which we termed EtgA). Both dehydrogenases are NAD-dependent. We show that the alcohol dehydrogenase EtgB exhibits a relatively high affinity for ethylene glycol, while also catalyzing the oxidation of various other alcohols. The aldehyde dehydrogenase EtgA efficiently converts glycolaldehyde into glycolate, enabling the subsequent conversion of glycolate to glyoxylate, which then enters central carbon metabolism via the BHAC. The cryo-EM structures of EtgA and EtgB, determined at 3.1 and 3.0 Å resolution, respectively, provide a detailed insight into the active site of these enzymes and a starting point for structure-guided mutagenesis. Furthermore, we show that the transcriptional regulator EtgR functions as an activator of ethylene glycol assimilation, and that evolved *P. denitrificans* strains are capable of faster growth on ethylene glycol due to increased activity of EtgB and EtgA. Our investigation demonstrates that NAD-dependent

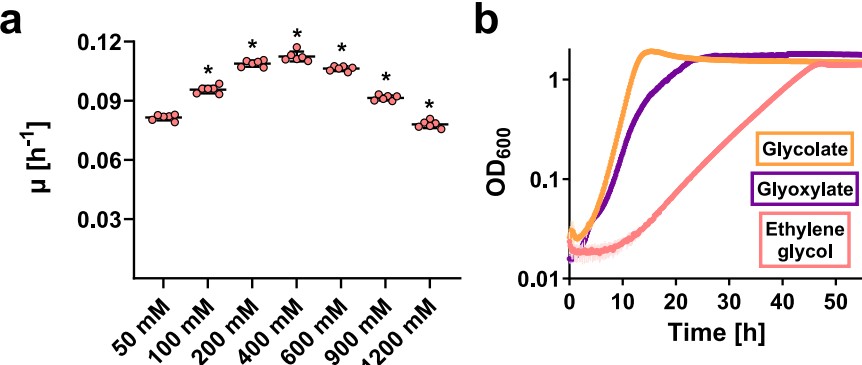

**Fig. 2 | Growth of *P. denitrificans* on ethylene glycol. a** Growth rates of *P. denitrificans* on different concentrations of ethylene glycol. Data are the mean ± s.d. of *n* = 6 independently grown cultures. Results were compared using an unpaired *t*-test with Welch's correction in GraphPad Prism 8.1.1. *: significantly different from growth rate with lower and higher concentrations of ethylene glycol (*p*-values in order of increasing ethylene glycol concentration: 0.0001; 0.0001; 0.017; 0.0009; 0.0001; 0.0001). **b** Growth curves of *P. denitrificans* on 60 mM glycolate (orange), 60 mM glyoxylate (purple), and 400 mM ethylene glycol (light red). Data are the mean ± s.d. of *n* = 6 independently grown cultures, with error bars shown in lighter colors. Source data are provided as a Source Data file.

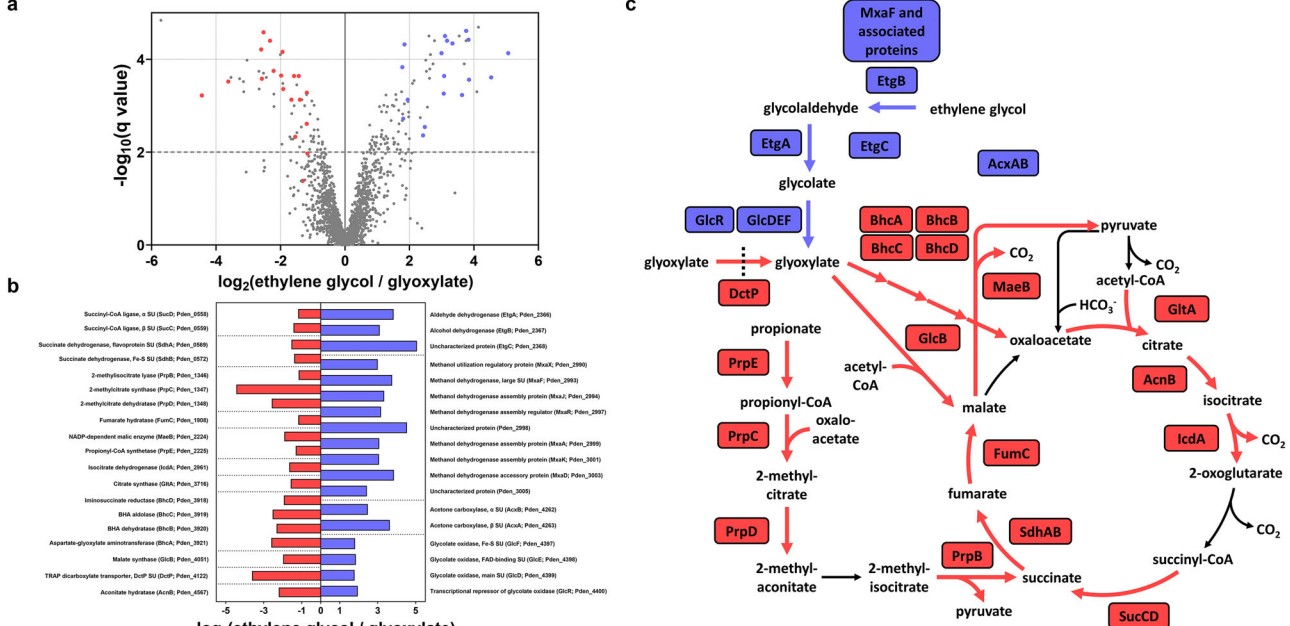

**Fig. 3 | Proteome analysis of *P. denitrificans* growing on ethylene glycol.**
**a** Analysis of the proteome of ethylene glycol-grown *P. denitrificans* compared to glyoxylate-grown *P. denitrificans* via mass spectrometry-based whole-cell shotgun proteomics. All proteins quantified by at least three unique peptides are shown, and the proteins involved in central carbon metabolism that showed the strongest decrease or increase in abundance are marked in red or blue in the volcano plot, respectively. **b** The log$_2$ fold change of these proteins, sorted by locus name (in brackets). **c** The role of these up- and downregulated proteins in central carbon metabolism of *P. denitrificans*. Source data are provided as a Source Data file.

dehydrogenases indeed enable efficient assimilation of ethylene glycol, and that similar gene clusters are widely distributed in bacteria. These findings elucidate key steps in the microbial transformation of ethylene glycol in the environment and pave the way for future applications of EtgB and EtgA in biocatalysis and PET upcycling.

## Results

### *P. denitrificans* can grow on ethylene glycol

To identify enzymes involved in ethylene glycol metabolism of *P. denitrificans*, we first tested its growth on minimal medium with this compound as the sole source of carbon and energy. The bacterium was growing on all tested concentrations of ethylene glycol, with an optimum growth rate at 400 mM (Fig. 2a). However, growth of *P. denitrificans* on 400 mM ethylene glycol was still considerably slower than on glycolate or glyoxylate (Fig. 2b). We observed an increased lag phase during growth on ethylene glycol, when compared to growth on glycolate or glyoxylate. The same phenotype was previously observed for *P. putida* JM37 and ascribed to the required induction of additional enzymes[26].

Next, we performed mass spectrometry-based whole-cell shotgun proteomics to analyze the proteome of *P. denitrificans* growing on either ethylene glycol or glyoxylate. With this approach, we aimed to identify the enzymes that are involved in ethylene glycol oxidation. We observed that the enzymes of the BHAC as well as several enzymes of the TCA cycle were downregulated on ethylene glycol, which is in line with the observed lower growth rate (Fig. 3). Notably, the *P. denitrificans* homologs of FucO and AldA, which catalyze the conversion of ethylene glycol to glycolate in an evolved strain of *E. coli*[29,30,39], were not strongly upregulated (Supplementary Fig. 1). Instead, a gene cluster comprising a NAD-dependent aldehyde dehydrogenase (Gene ID:

**Table 1 | Kinetic parameters of the enzymes characterized in this study**

| Enzyme | Substrate | $k_{cat}$ (s$^{-1}$) | App. $K_m$ (mM) | $k_{cat}/K_m$ (M$^{-1}$ s$^{-1}$) |
|---|---|---|---|---|
| EtgB WT | Ethanol[1] | 19.0 ± 0.9 | 1.17 ± 0.13 | (1.63 ± 0.20) * 10$^4$ |
| | Ethylene glycol[1] | 5.5 ± 0.1 | 84.92 ± 6.32 | (6.48 ± 0.50) * 10$^1$ |
| | Ethanol[2] | 23.5 ± 0.5 | 0.12 ± 0.01 | (2.01 ± 0.16) * 10$^5$ |
| | Ethylene glycol[2] | 7.2 ± 0.2 | 11.05 ± 1.32 | (6.51 ± 0.22) * 10$^2$ |
| | Propylene glycol[2] | 3.5 ± 0.1 | 59.46 ± 3.31 | (5.96 ± 0.10) * 10$^1$ |
| | n-propanol[2] | 18.7 ± 0.5 | 0.11 ± 0.01 | (1.76 ± 0.14) * 10$^5$ |
| | n-butanol[2] | 8.4 ± 0.1 | 0.04 ± 0.01 | (2.13 ± 0.14) * 10$^5$ |
| | Acetaldehyde[3] | 157.5 ± 2.9 | 0.04 ± 0.01 | (3.84 ± 0.24) * 10$^6$ |
| | Glycolaldehyde[3] | 50.1 ± 1.2 | 1.04 ± 0.06 | (4.83 ± 0.32) * 10$^4$ |
| EtgB T44S | Ethanol[2] | 3.2 ± 0.1 | 0.10 ± 0.02 | (3.32 ± 0.59) * 10$^4$ |
| | Ethylene glycol[2] | 0.7 ± 0.1 | 10.17 ± 0.36 | (7.02 ± 0.26) * 10$^1$ |
| | Propylene glycol[2] | 1.6 ± 0.1 | 27.68 ± 1.73 | (5.78 ± 0.07) * 10$^1$ |
| EtgB H47N | Ethanol[2] | 25.2 ± 0.4 | 0.23 ± 0.01 | (1.08 ± 0.05) * 10$^5$ |
| | Ethylene glycol[2] | 14.7 ± 0.3 | 53.69 ± 4.04 | (2.74 ± 0.06) * 10$^2$ |
| | Propylene glycol[2] | 13.2 ± 0.6 | 67.18 ± 8.90 | (1.97 ± 0.08) * 10$^2$ |
| EtgB T44S H47N | Ethanol[2] | 10.7 ± 0.4 | 0.10 ± 0.02 | (1.05 ± 0.21) * 10$^5$ |
| | Ethylene glycol[2] | 2.9 ± 0.1 | 11.51 ± 1.47 | (2.51 ± 0.05) * 10$^2$ |
| | Propylene glycol[2] | 6.5 ± 0.2 | 19.75 ± 2.98 | (3.30 ± 0.11) * 10$^2$ |
| EtgA | Acetaldehyde[4] | 1.3 ± 0.1 | 0.009 ± 0.001 | (1.35 ± 0.21) * 10$^5$ |
| | Glycolaldehyde[4] | 6.3 ± 0.2 | 0.12 ± 0.01 | (5.13 ± 0.49) * 10$^4$ |

Data are mean ± s.d., as determined from nonlinear fits of at least 21 data points with GraphPad Prism 8.1.1. Michaelis–Menten fits of enzyme kinetics are provided in Supplementary Fig. 4.

[1] Determined at 30 °C with 100 mM potassium phosphate buffer pH 7.5 and 2.4 mM NAD$^+$.

[2] Determined at 30 °C with 100 mM glycine-NaOH buffer pH 10 and 2.4 mM NAD$^+$.

[3] Determined at 30 °C with 100 mM potassium phosphate buffer pH 7.5 and 0.2 mM NADH.

[4] Determined at 30 °C with 100 mM potassium phosphate buffer pH 7.5 and 2.4 mM NAD$^+$.

Pden_2366; Uniprot ID: A1B4L2), a NAD-dependent alcohol dehydrogenase (Pden_2367; A1B4L3), and a protein of unknown function (Pden_2368; A1B4L4) was strongly upregulated. We hypothesized that these enzymes might be involved in the conversion of ethylene glycol to glycolate and therefore termed these genes *etgA*, *etgB*, and *etgC*, respectively. Furthermore, the three subunits of glycolate oxidase and the transcriptional regulator GlcR, which acts as an activator for the glycolate oxidase gene cluster[40], were strongly upregulated on ethylene glycol. This indicates that glycolate oxidase, and not glyoxylate reductase, as previously proposed[38], is responsible for the oxidation of glycolate to glyoxylate.

Moreover, the large subunit of PQQ-dependent methanol dehydrogenase (MxaF) was upregulated, together with several other proteins encoded by genes in the *mxa* gene cluster. We wondered whether this promiscuous enzyme[41,42] might be involved in ethylene glycol assimilation. Therefore, we generated two *P. denitrificans* Δ*mxaF* strains, in which the gene was replaced with a kanamycin resistance cassette in either the same or the opposite direction of transcription to exclude any polar effects. As expected, these gene deletion strains were unable to grow on methanol (Supplementary Fig. 2a). However, their growth rate on different concentrations of ethylene glycol was not significantly different from the growth rate of the WT (Supplementary Fig. 2b). We therefore concluded that the PQQ-dependent methanol dehydrogenase is not involved in ethylene glycol assimilation in *P. denitrificans*. Finally, acetone carboxylase (AcxAB) was also strongly upregulated on ethylene glycol. The production of this enzyme is known to be induced by acetone and isopropanol[43,44], while our results suggest that ethylene glycol or a downstream metabolite might also have an inducing effect in *P. denitrificans*.

### Structural and functional characterization of EtgA and EtgB

Next, we investigated the potential involvement of the *etg* gene cluster in ethylene glycol assimilation. We hypothesized that the alcohol dehydrogenase EtgB can convert ethylene glycol into glycolaldehyde, which is subsequently further converted into glycolate by the aldehyde dehydrogenase EtgA. To test this hypothesis, we generated *P. denitrificans* Δ*etgA* and Δ*etgB* strains, in which the respective gene was replaced with a kanamycin resistance cassette in either the same or the opposite direction of transcription to exclude any polar effects. These gene deletion strains grew WT-like on succinate (Supplementary Fig. 3a), but their growth rate on different concentrations of ethylene glycol was significantly decreased compared to the growth rate of the WT (Supplementary Fig. 3b). We therefore concluded that EtgB and EtgA are the key enzymes for the conversion of ethylene glycol into glycolate in *P. denitrificans*. In the absence of one of these enzymes, other alcohol or aldehyde dehydrogenases might be able to catalyze the respective conversion with low efficiency, resulting in the observed decreased growth rates. To further investigate EtgB and EtgA, we produced both enzymes in *E. coli* and subsequently purified them.

We performed enzyme assays confirming that EtgB is indeed an alcohol dehydrogenase that utilizes NAD$^+$ (NADH) as cofactor when converting alcohols to aldehydes (or vice versa). The kinetic parameters of EtgB with seven different substrates are listed in Table 1 (see also Supplementary Fig. 4). We found that this enzyme preferably accepts monoalcohols (ethanol, n-propanol, n-butanol) as substrates. No reaction was observed with methanol as substrate. The enzyme was found to catalyze the dehydrogenation of ethylene glycol, albeit with a $k_{cat}/K_m$ value that is 300-fold lower than the $k_{cat}/K_m$ for the conversion of ethanol. Nevertheless, when compared with NAD-dependent alcohol dehydrogenases characterized previously, EtgB shows favorable kinetic parameters with ethylene glycol as substrate (Supplementary Table 1). When produced in *E. coli* K-12, EtgB allowed growth on ethylene glycol (Supplementary Fig. 5). Furthermore, EtgB catalyzes the conversion of acetaldehyde to ethanol with an approximately 100-fold higher catalytic efficiency ($k_{cat}/K_m$) than the conversion of glycolaldehyde to ethylene glycol. To evaluate the suitability of the enzyme for in vitro biocatalysis, we also studied the effect of temperature and pH on the oxidation of ethylene glycol by EtgB. Among the tested

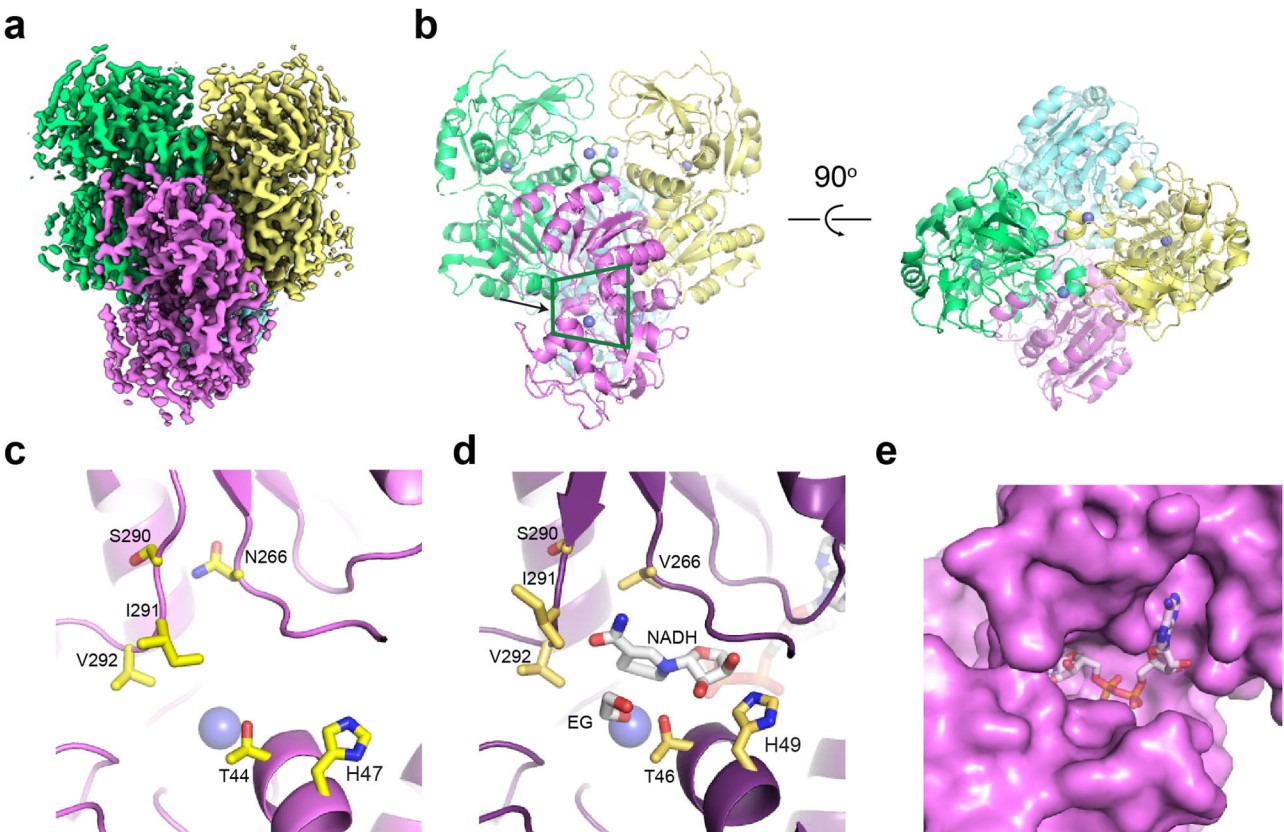

**Fig. 4 | Cryo-EM structure of EtgB. a** Cryo-EM map of EtgB with the four monomers shown in different colors. Fourth monomer is not visible in this view. **b** Left side: front view of the structure of EtgB shown in cartoon representation, with the four monomers shown in different colors. Zinc ions are shown in blue spheres. Green trapezium shows the view in (**c**) and (**d**). Right side: top view of EtgB, rotated 90° from front view. **c** Close up of the active site of EtgB with residues that line the cavity in yellow sticks. Zinc ion shown in blue sphere. **d** Active site of *P. aeruginosa* alcohol dehydrogenase[45] in complex with NADH and ethylene glycol (EG). Active site residues, NADH and EG shown in sticks. Zinc ion shown in blue sphere. **e** Surface representation of the EtgB active site, with the NADH molecule from *P. aeruginosa* alcohol dehydrogenase modeled into the cavity.

conditions, the optimal pH for the reduction direction was 6, while the optimal pH for the oxidation direction was 10. Enzyme activity in the oxidation direction increased with increasing temperature, consistent with a relatively high unfolding temperature of 78 °C (Supplementary Fig. 6).

To investigate whether there are specific structural features that facilitate the conversion of ethylene glycol, we obtained the cryo-EM structure of EtgB at 3.0 Å resolution (PDB 9FLZ; Supplementary Fig. 7). The tetrameric EtgB is a class II zinc-dependent medium-chain alcohol dehydrogenase (Fig. 4a, b, Supplementary Table 2). It is structurally similar to previously characterized alcohol dehydrogenases from bacteria and yeast[45–48] (Supplementary Table 3). Its active site contains a catalytic zinc ion and a proton-shuttling system including Thr44 and His47 (Fig. 4c), indicating that the reaction mechanism of EtgB runs analogous to other zinc-dependent alcohol dehydrogenases. In the active site of the structurally similar alcohol dehydrogenase from *Pseudomonas aeruginosa* (PDB 1LLU), the second hydroxyl group of ethylene glycol interferes with Thr46, which might explain why this diol is a poor substrate ($k_{cat}/K_m < 23\,M^{-1}\,s^{-1}$) for the enzyme (Fig. 4d, e)[45]. Therefore, we replaced the analogous Thr44 in EtgB with the smaller, isofunctional Ser to test for improved conversion of ethylene glycol and other diols. We also replaced His47 with Asn to probe the effect of this smaller amino acid on the active site and the proton-shuttling system. Finally, we created a Thr44Ser His47Asn double mutant to evaluate the combined effect of both substitutions. None of these mutations improved the catalytic efficiency for the dehydrogenation of ethylene glycol (Table 1). However, EtgB H47N and especially EtgB T44S H47N

showed improved catalytic efficiency with propylene glycol as substrate (ca. three-fold and ca. five-fold improved compared to EtgB WT, respectively). Therefore, the re-design of the proton-shuttling system with smaller amino acids has a positive effect on the conversion of this bulky three-carbon diol substrate. We hypothesize that the mutations may have opened up the active site to better accommodate propylene glycol, while ethylene glycol might not be optimally positioned in the larger active site cavity of these mutant variants of EtgB.

Subsequently, we aimed to confirm that EtgA is responsible for the oxidation of glycolaldehyde to glycolate. It was shown previously that this enzyme converts acetaldehyde to acetate[49]. Enzyme assays revealed that EtgA readily oxidizes glycolaldehyde, albeit with a 2.5-fold lower catalytic efficiency than acetaldehyde (Table 1; Supplementary Fig. 4). We also obtained the cryo-EM structure of EtgA at 3.1 Å resolution (PDB 9FM9; Supplementary Fig. 8), with some density missing in one of the four monomers (Fig. 5a, b). This homotetrameric enzyme is made up of a 'dimer-of-dimers', where the main dimer interfaces show Gibbs free energy (ΔG) values of −31.2 kcal/mol, indicative of positive protein affinity (Supplementary Fig. 9). In contrast, the inter-dimer contacts show energetically unfavorable ΔG values of +2.3 and +4.5 kcal/mol, consistent with weak contacts between the 'dimer-of-dimers', resulting in flexibility and a weak definition of the cryo-EM map. EtgA is structurally similar to other bacterial aldehyde dehydrogenases[50,51] (Supplementary Table 3) and has a canonical active site with catalytic Cys and Glu residues (Fig. 5c). The active site of EtgA is optimized for smaller aldehyde substrates, since Tyr464 takes the space that is occupied by Gly461 and the bulky product

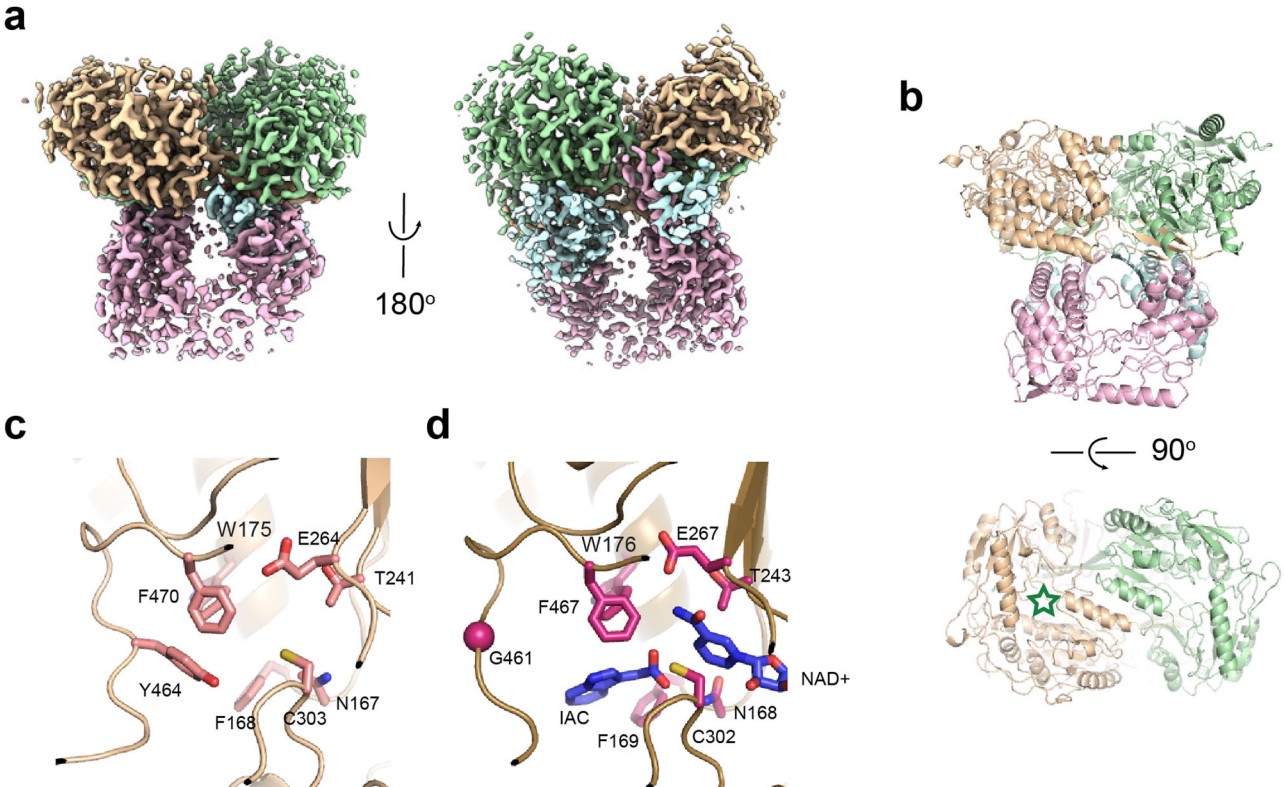

**Fig. 5 | Cryo-EM structure of EtgA. a** Left side: cryo-EM map of EtgA with the four monomers shown in different colors. The fourth monomer (light blue) is poorly visible in the cryo-EM map. Right side: same map rotated by 180° shows the absence of density for the fourth (light blue) monomer. **b** Front and top view of the structure of EtgA shown in cartoon representation, with the four monomers shown in different colors. The green star in the lower panel marks the location of the active site.

**c** Close up of the active site of EtgA with residues that line the putative substrate-binding cavity in light pink sticks. **d** Active site of *P. syringae* aldehyde dehydrogenase[50] in complex with NAD⁺ and indole-3 acetate (IAC). Active site residues, NAD⁺, and IAC are shown in magenta sticks. Note that Y464 in EtgA occupies the position of IAC in the *P. syringae* aldehyde dehydrogenase structure, where G461 is located at the same position.

molecule indole-3 acetate (IAC) in the structurally similar aldehyde dehydrogenase of *Pseudomonas syringae* (PDB 5IUW; Fig. 5d) [50]. When we introduced a Y464G mutation in EtgA, the resulting mutant enzyme did not catalyze the oxidation of acetaldehyde or glycolaldehyde anymore, supporting the hypothesis that the bulky tyrosine residue is key to the conversion of smaller aldehyde substrates.

### EtgC is not involved in aerobic ethylene glycol assimilation

To complete our investigation of the *etg* gene cluster, we sought to investigate the role of EtgC, a small protein of unknown function (12.8 kDa). Phylogenetic analysis revealed that EtgC and its homologs constitute a clade that forms a sister group to iron-sulfur (Fe-S) cluster assembly proteins such as IscA and SufA from *E. coli* and ErpA from *Haemophilus influenzae*. Several ferredoxins−electron transfer proteins with Fe-S clusters−that are present in *P. denitrificans* are less closely related to EtgC (Supplementary Fig. 10). EtgC and its homologs have four conserved cysteines, which commonly serve to coordinate (nascent) Fe-S clusters in ferredoxins or Fe-S cluster assembly proteins[52]. Comparative analysis of the predicted structure of EtgC demonstrated similarity to the structures of ErpA, SufA, and IscA (Supplementary Table 3). It therefore seemed plausible that EtgC might also contain a Fe-S cluster. Next, we purified the heterologously produced protein and analyzed it via inductively coupled plasma optical emission spectroscopy (ICP-OES) to investigate the presence of iron atoms. Notably, this analysis revealed that neither iron nor cobalt, copper, manganese, or zinc were present in EtgC in stoichiometric amounts (Supplementary Table 4). We therefore concluded that EtgC either does not contain a Fe-S cluster, or that the Fe atom(s) were not properly inserted into the heterologously produced protein.

To investigate a possible role of EtgC in ethylene glycol assimilation, we generated two *P. denitrificans* Δ*etgC* strains, in which the gene was replaced with a kanamycin resistance cassette in either the same or the opposite direction of transcription to exclude any polar effects. The growth rate of these gene deletion strains was not significantly different from the growth rate of the WT on different concentrations of ethylene glycol (Supplementary Fig. 11a). We also compared the growth of the WT and Δ*etgC* strains in the absence of oxygen with nitrate as terminal electron acceptor. Growth of Δ*etgC* was slightly, but significantly, impaired under these conditions (Supplementary Fig. 11b, c). EtgC is therefore not involved in aerobic ethylene glycol assimilation, while its potential function in anaerobic ethylene glycol metabolism may deserve further investigation.

### EtgR acts as activator of the *etg* gene cluster

The gene Pden_2365 (Uniprot ID: A1B4L1), adjacent to the *etgABC* gene cluster, is annotated as Fis family transcription factor. We hypothesized that this gene (henceforth called *etgR*) encodes for a regulator of the *etg* gene cluster. To test this hypothesis, we generated two gene deletion strains in which *etgR* was replaced with a kanamycin resistance cassette in either the same or the opposite direction of transcription. Growth assays on different carbon sources showed that Δ*etgR* grew on succinate just like the WT, but was unable to grow on ethylene glycol and barely grew on ethanol. Growth rates on other alcohols were decreased to a lesser degree (Fig. 6a). We therefore concluded that deletion of *etgR* abolishes expression of the *etg* gene cluster, indicating that EtgR acts as an activator. These results suggest that there are no suitable alternative enzymes for the oxidation of ethylene glycol in *P. denitrificans*, even though several other genes

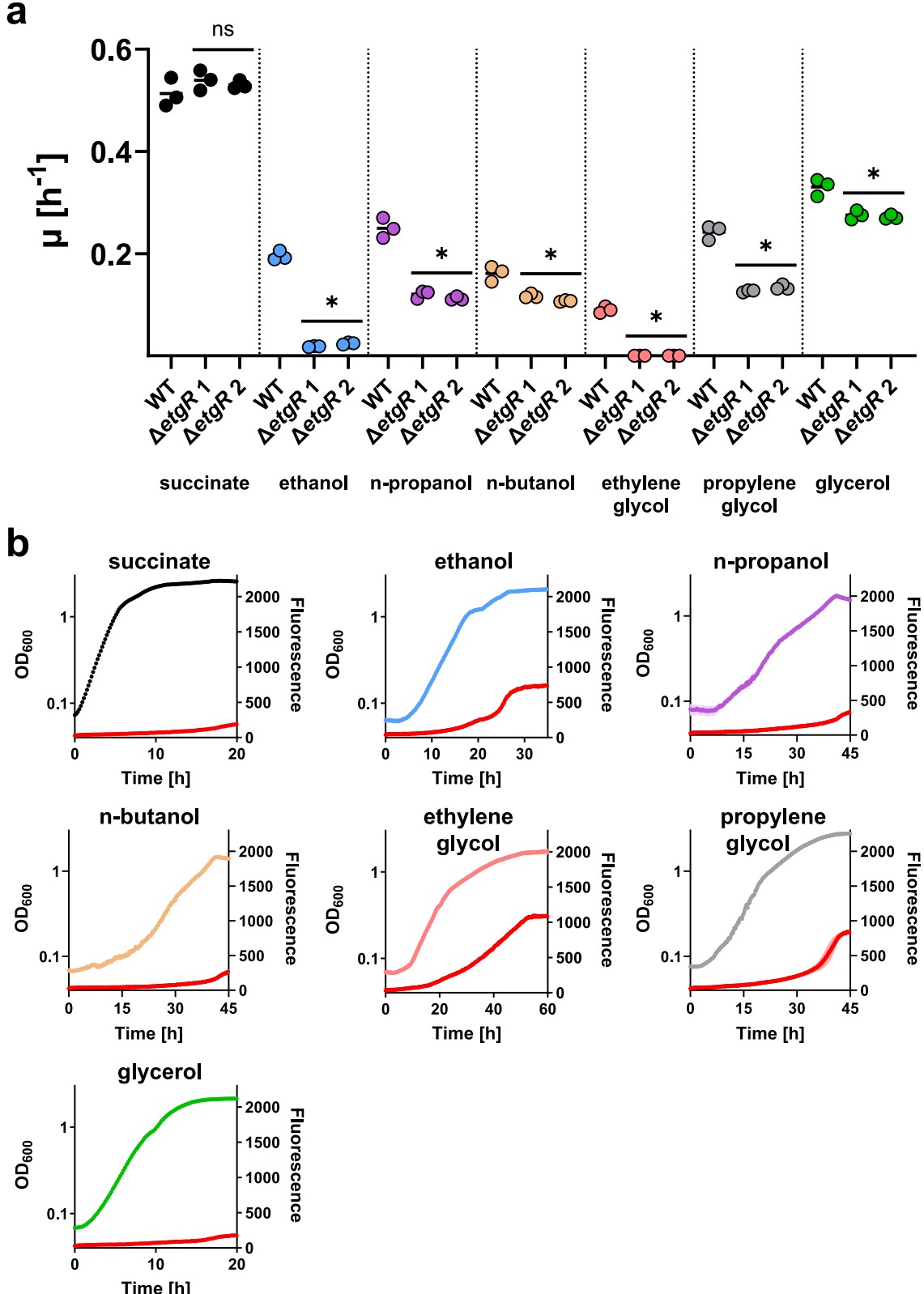

**Fig. 6 | Characterization of *P. denitrificans* Δ*etgR* and promoter reporter strains on different carbon sources. a** Growth rates of *P. denitrificans* WT and Δ*etgR* strains on succinate and different alcohols as sole sources of carbon and energy. $n = 3$ independent cultures per strain and carbon source. Results were compared using an unpaired *t*-test with Welch's correction in GraphPad Prism 8.1.1. ns: not significantly different from WT; *: significantly different from WT (*p*-values for succinate: 0.2643, 0.4089; ethanol: 0.0008, 0.0007; n-propanol: 0.0032, 0.0054;

n-butanol: 0.0285, 0.0237; ethylene glycol: 0.0017, 0.0017; propylene glycol: 0.0042, 0.0026; glycerol: 0.0143, 0.0194). **b** Growth and fluorescence (red) of *P. denitrificans* with pTE714-P*etg* on different carbon sources. Data are the mean ± s.d. of $n = 3$ independently grown cultures, with error bars shown in lighter colors. Growth and fluorescence of negative control strains are shown in Supplementary Fig. 15. Source data are provided as a Source Data file.

**Table 2 | Variants found in genome resequencing data of three isolated strains of *P. denitrificans* after adaptive laboratory evolution on ethylene glycol**

| Strain | Gene ID | Mutation | Annotation | Position | Protein |
|---|---|---|---|---|---|
| Evo 1 | Pden_0444 | A → G | D175G | C1: 413,287 | Uncharacterized protein |
| | Pden_2365 | T → C | V266A | C1: 2,365,949 | EtgR (Transcriptional regulator, Fis family) |
| | IG region of Pden_3941/42 | C → G | | C2: 1,079,285 | |
| Evo 2 | Pden_0303 | T → C | I24T | C1: 284,506 | Transmembrane pair domain protein |
| | Pden_2365 | A → C | D193A | C1: 2,365,730 | EtgR (Transcriptional regulator, Fis family) |
| | Pden_3966 | A → G | M139V | C2: 1,104,667 | FeS assembly ATPase SufC |
| Evo 3 | IG region of Pden_3259/60 | C → A | | C2: 422,441 | |
| | Pden_2334–Pden_2415 | region with strongly increased coverage | | C1: 2,331,893–2,422,134 | |

*IG* intergenic, *C1* chromosome 1, *C2* chromosome 2.

encoding for NAD-dependent alcohol dehydrogenases are present in its genome, and the PQQ-dependent alcohol dehydrogenase MxaF was upregulated on ethylene glycol. In contrast, there must be enzymes that mediate the oxidation of other alcohols, enabling decreased growth of Δ*etgR*.

Next, we investigated under which conditions EtgR activates expression of the *etg* gene cluster. To this end, we created a promoter probe vector in which the gene encoding for mCherry is under the control of $P_{etg}$, the putative promoter region of the *etg* gene cluster. Subsequently, we introduced this vector into *P. denitrificans* and grew the resulting promoter reporter strain on succinate and different alcohols. The $P_{etg}$ promoter reporter vector exhibited strong fluorescence on ethylene glycol, and to a lesser degree on ethanol and propylene glycol. Only low fluorescence was recorded on n-propanol, n-butanol, and glycerol (Fig. 6b). These results are consistent with the decreased growth rates of Δ*etgR* and suggest that different alcohols (or their downstream metabolites) interact to varying extents with the transcriptional regulator EtgR to activate the $P_{etg}$ promoter. Furthermore, we evaluated growth of the Δ*etgR* gene deletion strains with the promoter probe vector on minimal medium with 10 mM succinate and 60 mM ethylene glycol (Supplementary Fig. 12). Their limited growth suggests that only succinate was used for growth, and the low background fluorescence as well as the lack of additional fluorescence in the presence of ethylene glycol further confirm the characterization of EtgR as an activator.

### Adaptive laboratory evolution of *P. denitrificans* on ethylene glycol results in faster growth due to increased production levels of EtgB and EtgA

Since *P. denitrificans* grows faster on glycolate than ethylene glycol (Fig. 2b), we hypothesized that growth rate on the diol is limited by either enzyme kinetic parameters or production levels of EtgB and/or EtgA. To investigate this hypothesis, we performed adaptive laboratory evolution by conducting ten controlled transfers of *P. denitrificans* on minimal medium containing 200 mM ethylene glycol. Subsequently, we isolated three evolved strains, which will henceforth be referred to as Evo 1, Evo 2, and Evo 3. Notably, these evolved strains exhibited significantly improved growth rates on ethylene glycol compared to the parental WT strain (Supplementary Fig. 13a). To understand the molecular basis of this phenotype, we characterized the three evolved strains via genome resequencing and proteome analysis. All evolved strains were found to have variants in their chromosomes (Table 2).

Interestingly, two evolved strains carry non-silent mutations in *etgR*, which might alter the expression level of the *etgABC* genes. The third evolved strain shows strongly increased coverage of a ~90,000 bp region including the *etg* gene cluster, suggesting that duplication of this chromosomal region might have occurred. Proteome analysis of the evolved strains revealed that the three enzymes

EtgA, EtgB, and EtgC, and also the regulator EtgR, were upregulated more strongly, compared to *P. denitrificans* WT growing on ethylene glycol (Supplementary Fig. 1). In contrast, other alcohol and aldehyde dehydrogenases (MxaF, FucO, AldA) were not upregulated more strongly in the evolved strains, confirming that EtgB and EtgA are indeed the preferred enzymes for the conversion of ethylene glycol into glycolate. Mutations in the coding sequences of *etgB* or *etgA* were not observed in any of the evolved strains; therefore, the kinetic parameters of the encoded enzymes must be unchanged compared to the WT.

Next, we investigated the effect of the mutations on the expression of the *etg* gene cluster by implementing the $P_{etg}$ promoter reporter vector in the evolved strains. Like in the WT, fluorescence was strongly induced during growth on ethanol or ethylene glycol, but the level of fluorescence varied considerably between the WT and evolved strains (Supplementary Fig. 14), confirming that the mutations in *etgR* result in altered induction capabilities. As an additional line of evidence, we performed enzyme assays for EtgB and EtgA in cell-free extracts of the WT and evolved strains grown on either succinate or ethylene glycol. Enzyme activities of the alcohol or aldehyde dehydrogenases could not be measured in cell-free extracts of succinate-grown cells. However, strong enzyme activities of EtgB and EtgA were detectable in ethylene glycol-grown cells (Supplementary Fig. 13b, c). Notably, EtgB and/or EtgA enzyme activities in all evolved strains were consistently at least twofold higher than those in the WT. This supports the hypothesis that either mutated EtgR variants (in Evo 1 and 2) or gene duplications (in Evo 3) result in increased enzyme levels and therefore higher metabolic flux from ethylene glycol to glycolate, which directly results in increased growth rates of the evolved strains.

### Homologs of the *etg* genes are widely distributed among bacteria

Finally, we aimed to determine whether the *etg* genes—and therefore the ability to grow on ethylene glycol via NAD-dependent dehydrogenases—are present in other bacteria as well. Our search for EtgR/A/B/C revealed homologs of all four proteins in 3793 bacterial strains, which mostly belong to the classes *Alpha-* and *Gammaproteobacteria*, *Actinomycetes*, *Bacilli*, and *Bacteroidia* (Fig. 7a). Strains with homologs of all four proteins are distributed across the bacterial tree of life and also occur in other classes, for example *Myxococcia* and *Acidobacteriae* (Fig. 7b, Supplementary Data 1). The arrangement of the four genes as *etgRABC* is not conserved among this large number of strains (Supplementary Data 2), but there are many examples with an *etgABC* gene cluster adjacent to either *etgR* or a gene encoding for another transcriptional regulator. Other strains have a gene cluster consisting of only *etgAC* and either *etgR* or a gene encoding for another transcriptional regulator (Fig. 7c), with an *etgB* homolog located elsewhere in the genome (Supplementary Data 2). In summary, the *etg* genes, but not necessarily their arrangement as an *etgRABC* gene cluster, are

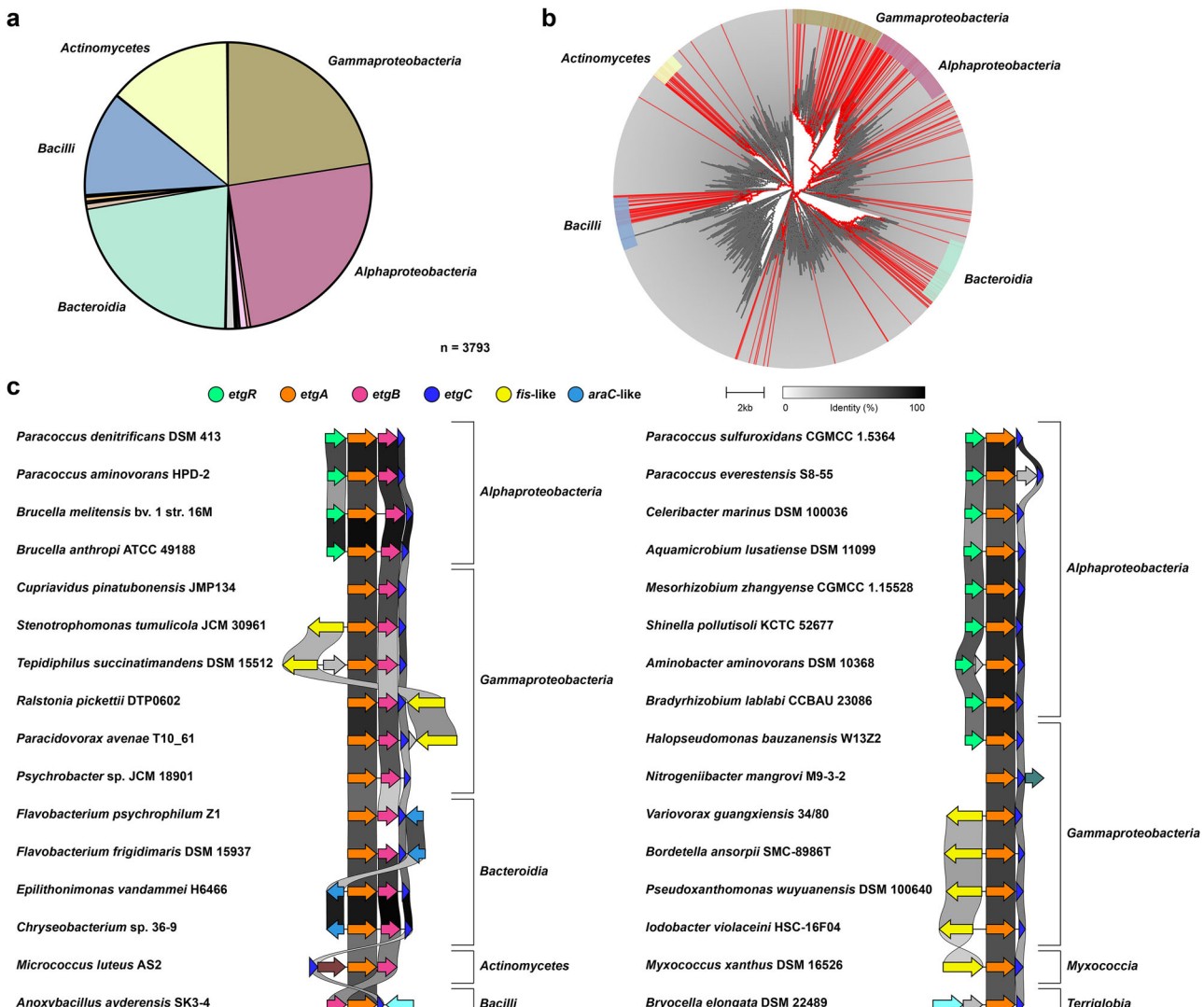

**Fig. 7 | Variants of the *etg* gene cluster are widely distributed among bacteria.** **a** Phylogenetic distribution of 3793 bacterial strains with homologs of EtgR, EtgA, EtgB, and EtgC in their genomes. **b** Genome-based phylogenetic tree of bacterial families; families containing strains with EtgR/A/B/C homologs are marked in red. The five bacterial classes named in (**a**) are highlighted. **a**, **b** Taxonomy is based on GTDB[91,92] (release R95; https://gtdb.ecogenomic.org/). **c** Selected examples of *etg* gene clusters from diverse bacteria. Left: gene clusters with *etgA*, *etgB*, *etgC*, and *etgR* or another gene encoding for a transcriptional regulator; right: gene clusters with *etgA*, *etgC*, and *etgR* or another gene encoding for a transcriptional regulator. Pairwise identity of homologous proteins is denoted in shades of gray.

widely distributed among bacteria and may play a common role in the microbial assimilation of ethylene glycol.

## Discussion

Ethylene glycol is widely used as a monomer of PET, as a solvent, as an antifreeze agent, and as a next-generation feedstock in microbial biotechnology. Therefore, the investigation of ethylene glycol-converting enzymes and metabolic routes is highly relevant to develop efficient bioprocesses for the upcycling of this diol. Here we confirmed the ability of the soil bacterium *P. denitrificans* to grow on ethylene glycol as the sole source of carbon and energy. Using a comparative proteomics approach, we observed a notable upregulation in the expression of a gene cluster on ethylene glycol. Subsequently, we characterized this gene cluster, which includes *etgR*, *etgA*, *etgB*, and *etgC*, in detail. We showed that the NAD-dependent alcohol dehydrogenase EtgB catalyzes the conversion of ethylene glycol to glycolaldehyde with high efficiency in comparison to other alcohol dehydrogenases. Next, glycolaldehyde is further oxidized by the aldehyde dehydrogenase EtgA. The transcriptional regulator EtgR activates expression of the *etg* gene cluster in the presence of ethylene

glycol and other alcohols. We furthermore demonstrate that increased enzyme activities of EtgB and EtgA result in faster growth of evolved strains on ethylene glycol, and that homologs of the *etg* genes are widely distributed among bacteria.

While the oxidation of ethylene glycol by PQQ- and MFT-dependent alcohol dehydrogenases was characterized previously, it was unclear whether NAD-dependent enzymes can also efficiently mediate this conversion. FucO from *E. coli* is NAD-dependent, but this iron-containing group III alcohol dehydrogenase[53] mainly functions in the anaerobic reduction of lactaldehyde to propylene glycol[39], and only mediates the oxidation of ethylene glycol in evolved strains of this model organism[29]. Gox0313 from *Gluconobacter oxydans* is NAD-dependent and oxidizes ethylene glycol in vitro[54], but growth on this diol is rather mediated by a membrane-bound PQQ-dependent alcohol dehydrogenase[55]. While the PQQ-dependent methanol dehydrogenase of *P. denitrificans* is upregulated during growth on ethylene glycol, our work demonstrates that Δ*mxaF* strains still grow WT-like on ethylene glycol. In contrast, the growth rate of Δ*etgB* strains on ethylene glycol is strongly decreased. Therefore, EtgB represents an NAD-dependent alcohol dehydrogenase that naturally mediates the growth of its

microbial host on ethylene glycol. The relatively good growth rate of *P. denitrificans* on ethylene glycol with only NAD-dependent enzymes can possibly be ascribed to high production of EtgB and a low $K_m$ of EtgA for glycolaldehyde. While the kinetic parameters of EtgB for ethylene glycol conversion are less favorable, they are still among the best reported so far. This makes this enzyme a promising candidate for in vitro biocatalysis, especially when combined with a suitable NAD$^+$ regeneration system. Furthermore, it is likely that EtgB can be engineered to optimize its catalytic performance, based on the structural information reported in this study.

The heterologous production of PQQ-dependent enzymes is difficult, since they are often membrane-bound, localized in the periplasm, and might require the co-production of up to six enzymes for PQQ biosynthesis. Therefore, NAD-dependent dehydrogenases are frequently preferred in metabolic engineering approaches due to their localization in the cytosol and the ubiquity of NAD biosynthesis. Recently, the NAD-dependent alcohol dehydrogenases FucO and Gox0313 were applied to implement synthetic pathways for the conversion of ethylene glycol into 2,4-dihydroxybutyrate[56] or aromatic amino acids[11] in *E. coli*. EtgB is a promising enzyme for similar in vivo engineering projects in the future, especially since we demonstrated the growth of *E. coli* K-12 on ethylene glycol when equipped with EtgB. As an alternative to the heterologous expression of *etgB* in a host bacterium, *P. denitrificans* itself could be engineered for the production of value-added molecules from ethylene glycol, since genetic tools for this bacterium[57,58] and its close relative *Paracoccus pantotrophus*[59] are available. Besides applications of the enzymes EtgB and EtgA, the transcriptional regulator EtgR is a promising candidate for biosensor development. Future work should focus on its in vitro characterization, followed by the engineering of dynamic and robust sensor modules. A biosensor that rapidly responds to ethylene glycol will enable high-throughput screening of improved strains for both ethylene glycol production[60–62] and PET degradation[9,13,14].

In summary, our results provide comprehensive insights into the assimilation of ethylene glycol by *P. denitrificans* and demonstrate that NAD-dependent dehydrogenases enable efficient conversion of this diol. Therefore, this study paves the way toward applications of the involved proteins in biocatalysis and metabolic engineering for sustainable production with PET-derived ethylene glycol as feedstock.

## Methods
### Chemicals and reagents
Unless otherwise stated, all chemicals and reagents were acquired from Sigma-Aldrich (St. Louis, USA) or VWR International BV (Amsterdam, the Netherlands) and were of the highest purity available.

### Strains, media and cultivation conditions
All strains used in this study are listed in Supplementary Table 5. *Escherichia coli* DH5α (for genetic work), ST18[63] (for plasmid conjugation to *P. denitrificans*), BL21 AI (for protein production) and K-12 W3110 (for heterologous expression of *etgB*) were grown at 37 °C in lysogeny broth[64].

*Paracoccus denitrificans* DSM 413[65] and its derivatives were grown at 30 °C in lysogeny broth or in mineral salt medium with TE3-Zn trace elements[66] supplemented with various carbon sources. To monitor growth, the $OD_{600}$ of culture samples was determined on a photospectrometer (Merck Chemicals GmbH, Darmstadt, Germany).

### Vector construction
All plasmids used in this study are listed in Supplementary Table 6. To create a plasmid for heterologous overexpression of *etgC* in *E. coli*, this gene (Pden_2368) was cloned into the standard expression vector pET16b. To this end, the gene was amplified from genomic DNA of *P. denitrificans* DSM 413 with the primers provided in Supplementary Table 7. The resulting PCR product was digested with suitable

restriction endonucleases (Thermo Fisher Scientific, Waltham, USA) as given in Supplementary Table 7 and ligated into the expression vector pET16b that had been digested with the same enzymes to create a vector for heterologous production of EtgC. The genes encoding for EtgA (Pden_2366) and EtgB (Pden_2367) were cloned into pET16b using the same approach.

To create a plasmid for heterologous expression of *etgB* in *E. coli*, this gene (Pden_2367) was cloned into the expression vector pZ-ASS[67]. To this end, the gene was amplified from genomic DNA of *P. denitrificans* DSM 413 with the primers provided in Supplementary Table 7. Using Gibson assembly, the resulting PCR product was inserted into the backbone of pZ-ASS that had been amplified from the plasmid pZ-ASS with the primers provided in Supplementary Table 7 to create a vector for heterologous expression of *etgB*.

To create constructs for gene deletion in *P. denitrificans*, the upstream and downstream flanking regions of the genes to be deleted from *P. denitrificans* DSM 413 were cloned into the gene deletion vector pREDSIX[58]. To this end, the flanking regions were amplified from genomic DNA of *P. denitrificans* DSM 413 with the primers given in Supplementary Table 7. The resulting PCR products were used to perform Gibson assembly with the vector pREDSIX, which had been digested with *Mfe*I. Subsequently, the resulting vector was digested with *Nde*I (or *Kpn*I), and a kanamycin resistance cassette, which had been cut out of the vector pRGD-Kan with *Nde*I (or *Kpn*I), was ligated into the cut site to generate the final vectors for gene deletion.

To create a reporter plasmid for *P. denitrificans*, the intergenic region between *etgR/etgA* (Pden_2365/Pden_2366) was cloned into the promoter probe vector pTE714[40]. The region was amplified from genomic DNA of *P. denitrificans* DSM 413 with the primers provided in Supplementary Table 7. The resulting PCR product was digested with suitable restriction endonucleases (Thermo Fisher Scientific, Waltham, USA) as given in Supplementary Table 7 and ligated into likewise digested pTE714. Successful cloning of all desired constructs was verified by Sanger sequencing (Macrogen, Amsterdam, the Netherlands).

### Production and purification of recombinant proteins
For heterologous overproduction of EtgA, EtgB, and EtgC, the corresponding plasmid encoding for the respective protein was first transformed into chemically competent *E. coli* BL21 AI cells. The cells were then grown on LB agar plates containing 100 µg mL$^{-1}$ ampicillin at 37 °C overnight. A starter culture in selective LB medium was inoculated from a single colony on the next day and left to grow overnight at 37 °C in a shaking incubator at 180 rpm. The starter culture was used on the next day to inoculate a production culture in selective TB medium in a 1:100 dilution. The production culture was grown at 37 °C in a shaking incubator at 180 rpm to an $OD_{600}$ of 0.5 to 0.7, induced with 0.5 mM IPTG and 0.2% L-arabinose and subsequently grown overnight at 18 °C in a shaking incubator at 180 rpm. Cells were harvested at 6000 × *g* for 15 min at 4 °C, and cell pellets were stored at −20 °C until purification of enzymes.

Cell pellets were resuspended in twice their volume of buffer A (EtgA/EtgB: 300 mM sodium chloride, 25 mM potassium phosphate pH 8.0, 15 mM imidazole, 1 mM β-mercaptoethanol, and one tablet of SIGMAFAST™ protease inhibitor cocktail, EDTA-free, per L; EtgC: identical, but pH 8.5). The cell suspension was treated with a Sonopuls GM200 sonicator (BANDELIN electronic GmbH & Co. KG, Berlin, Germany) at an amplitude of 35% in order to lyse the cells and subsequently centrifuged at 50,000 × *g* and 4 °C for 1 h. The filtered supernatant (0.45 µm filter; Sarstedt, Nümbrecht, Germany) was loaded onto a Ni-NTA column (Cytiva, Medemblik, the Netherlands) connected to an ÄKTA Start FPLC system (Cytiva, Medemblik, the Netherlands), which had previously been equilibrated with 5 column volumes of buffer A. The column was washed with 20 column volumes of buffer A and 5 column volumes of 85% buffer A and 15% buffer B, and

the His-tagged protein was eluted with buffer B (buffer A with 500 mM imidazole). The eluate was desalted using PD-10 desalting columns (Cytiva, Medemblik, the Netherlands) and buffer C (EtgA/EtgB: 100 mM sodium chloride, 25 mM potassium phosphate pH 8.0, 1 mM DTT; EtgC: identical, but pH 8.5). Purified proteins in buffer C were subsequently used for downstream experiments.

Before structure determination, affinity purification was followed by purification on a size exclusion column (Superdex 200 Increase 10/300 GL; Cytiva, Medemblik, the Netherlands) connected to an ÄKTA Pure system (Cytiva, Medemblik, the Netherlands) using buffer C. 500 μL concentrated protein solution was injected, and flow was kept constant at 0.75 mL min$^{-1}$. Elution fractions containing pure protein were determined via SDS-PAGE analysis[68] on 12.5% gels.

### Enzyme activity assays
For all enzyme assays, the reduction of NAD$^+$ or oxidation of NADH was followed at 340 nm on a Cary 60 UV-Vis photospectrometer (Agilent, Santa Clara, USA) in quartz cuvettes with a path length of 1 mm (Hellma Optik, Jena, Germany).

The enzyme assays to determine the kinetic parameters of EtgB with different alcohols or aldehydes as substrates were performed at 30 °C in a total volume of 300 μL. The reaction mixture contained 100 mM glycine-NaOH buffer pH 10, 2.4 mM NAD$^+$, different amounts of the respective substrates, and 120 nM EtgB in the alcohol oxidation assays and 100 mM potassium phosphate buffer pH 7.5, 0.2 mM NADH, different amounts of the respective substrates, and 60 nM EtgB in the aldehyde reduction assays. The enzyme assays to determine the kinetic parameters of EtgA with different aldehydes as substrates were performed at 30 °C in a total volume of 300 μL. The reaction mixture contained 100 mM potassium phosphate buffer pH 7.5, 2.4 mM NAD$^+$, different amounts of the respective substrates, and 69 nM EtgA.

### Enzyme activity assays in *P. denitrificans* cell-free extracts
*P. denitrificans* cultures were collected during mid-exponential phase (OD$_{600}$ of 0.5–0.7), resuspended in ice-cold 100 mM potassium phosphate buffer (pH 7.2), and lysed by sonication. Cell debris was separated by centrifugation at 35,000 × $g$ and 4 °C for 1 h. Total protein concentrations of the resulting cell-free extracts were determined by Bradford assay[69] using bovine serum albumin as standard. The assays for the activity of EtgA and EtgB were performed as described above.

### Protein thermal stability measurement
For the purified proteins EtgA, EtgB, and EtgC, the thermal stability was measured using a Tycho NT.6 instrument (NanoTemper Technologies GmbH, Munich, Germany). A constant heating rate of 30 °C per minute was applied to the samples, heating from 35 °C to 95 °C.

### Electron microscopy data collection
Cryo-electron microscopy data was collected on a Titan Krios operating at 300 kV (Thermo Fisher Scientific, Eindhoven, the Netherlands) equipped with a BioQuantum energy filter using a slit width of 20 eV and K3 direct electron detector (Gatan, Inc, Pleasanton, CA, USA) at the Netherlands Centre for Electron Nanoscopy (NeCEN). Movies were recorded with EPU software (Thermo Fisher Scientific) operating in electron counting mode and using aberration free image shift (AFIS) at a nominal magnification of 105.000 x (0.836 Å/pixel). The defocus range was set from −0.8 to −2.2 μm and the total accumulated dose to 50 electrons/Å$^2$ (1 e/Å$^2$ per frame).

### Electron microscopy data processing
Cryo-electron microscopy data were processed using RELION-5.0[70]. Movies were motion corrected and dose-weighted using MotionCor2[71] and CTF estimated using CTFFIND-4. Particles were identified with Topaz[72] using the standard pre-trained neural network. After extraction, particles were submitted to several rounds of 2D classification, ab-initio

model generation and 3D classification. Final 3D reconstructions were obtained after two iterative rounds of 3D refinement, CTF refinement (beam-tilt and aberration correction, anisotropic magnification estimation, and per-particle defocus and per-micrograph astigmatism correction) and Bayesian polishing. The final reconstruction was sharpened and locally filtered using RELION post-processing and yielded a final average resolution of 3.0 Å for EtgB and 3.1 Å for EtgA.

### Model building and refinement
Models for monomeric EtgA and EtgB were derived from AlphaFold 2[73] and superimposed onto the crystal structures of tetrameric *Pseudomonas syringae* aldehyde dehydrogenase[50] (PDB code 5IUW) or tetrameric *Pseudomonas aeruginosa* alcohol dehydrogenase[45] (PDB code 1LLU), respectively. Models were manually adjusted in Coot[74] and refined in Phenix[75]. Figures were prepared using ChimeraX[76] and PyMOL[77]. Interface energy values were calculated using the EBI PISA server[78] (https://www.ebi.ac.uk/pdbe/pisa).

### Metal content determination of EtgC
Inductively coupled plasma optical emission spectroscopy (ICP-OES) was used for the metal content quantification of EtgC. 0.715 mg of purified EtgC was dissolved in 0.5 mL trace metal grade concentrated nitric acid overnight at room temperature, followed by boiling for 2 h at 70 °C. Samples were diluted to a final volume of 8.5 mL in double-distilled water. Sample analysis was conducted using a 720/725 ICP-OES device (Agilent, Santa Clara, USA), with the following wavelengths used for metal determination: iron (λ = 238.204 nm), cobalt (λ = 228.615 nm), copper (λ = 327.395 nm), manganese (λ = 257.610 nm) and zinc (λ = 213.857 nm). The device was operated with ICP Expert v4.1.0 software (Agilent, Santa Clara, USA), and an ICP multi-element standard solution IV (Merck Chemicals GmbH, Darmstadt, Germany) was used as the standard for the quantification of the metals.

### Genetic modification of *P. denitrificans*
Transfer of replicative plasmids into *P. denitrificans* was performed via conjugation using *E. coli* ST18 as the donor strain. As described previously[36], *E. coli* ST18 containing the plasmid to be conjugated was grown on LB agar plates containing 100 μg mL$^{-1}$ ampicillin, 50 μg mL$^{-1}$ kanamycin and 50 μg mL$^{-1}$ aminolevulinic acid at 37 °C overnight. A culture in selective LB medium was inoculated the next day and left to grow overnight at 37 °C. The culture was diluted the next morning to an OD$_{600}$ of 0.1. A culture of *P. denitrificans* DSM 413 in LB medium was inoculated from a glycerol stock and grown at 30 °C. ST18 cultures were collected at an OD$_{600}$ of around 0.7, and the *P. denitrificans* culture was collected at an OD$_{600}$ of about 1.3. All cell pellets were washed once with sterile 10 mM MgSO$_4$ and resuspended to an OD$_{600}$ of approximately 10 in sterile 10 mM MgSO$_4$. Suspensions of ST18 cells and *P. denitrificans* cells were mixed in a 2:1 ratio and spotted on minimal medium agar plates without any carbon source. Plates were incubated at 30 °C overnight. The next morning, spots were removed from the plates, resuspended in LB medium and plated on selective LB agar plates. Selection of conjugants was performed at 30 °C on LB plates containing 1 μg mL$^{-1}$ tetracycline. Successful transfer of plasmids into *P. denitrificans* was verified by colony PCR.

Transfer of gene deletion plasmids into *P. denitrificans* was performed in the same way. Selection of conjugants was performed at 30 °C on LB agar plates containing 25 μg mL$^{-1}$ kanamycin. The respective gene deletion was verified by colony PCR and DNA sequencing and the deletion strain was propagated in selective LB medium.

### High-throughput growth and fluorescence assays with *P. denitrificans* and *E. coli*
Cultures of *P. denitrificans* DSM 413 WT and its derivatives were pre-grown at 30 °C in LB medium containing 25 μg mL$^{-1}$ kanamycin or

$1\,\mu g\,mL^{-1}$ tetracycline, when appropriate. Cells were harvested, washed once with minimal medium containing no carbon source and used to inoculate growth cultures of $180\,\mu L$ minimal medium containing an appropriate carbon source as well as $25\,\mu g\,mL^{-1}$ kanamycin or $1\,\mu g\,mL^{-1}$ tetracycline, when appropriate. Growth and fluorescence in 96-well plates (Thermo Fisher Scientific, Waltham, USA) were monitored at $30\,°C$ at 600 nm in a Tecan Infinite M200Pro plate reader (Tecan, Männedorf, Switzerland). Fluorescence of mCherry was measured at an emission wavelength of 610 nm after excitation at 575 nm, with a fixed gain of 100.

Cultures of *E. coli* K-12 W3110 and its derivatives were pre-grown at $37\,°C$ in LB medium containing $50\,\mu g\,mL^{-1}$ streptomycin, when appropriate. Cells were harvested, washed once with M9 minimal medium containing no carbon source and used to inoculate growth cultures of $180\,\mu L$ M9 minimal medium containing an appropriate carbon source as well as $50\,\mu g\,mL^{-1}$ streptomycin, when appropriate. Growth in 96-well plates (Thermo Fisher Scientific, Waltham, USA) was monitored at $37\,°C$ at 600 nm in a Tecan Infinite M200Pro plate reader (Tecan, Männedorf, Switzerland).

### Anaerobic growth of *P. denitrificans*

*P. denitrificans* WT and Δ*etgC* were grown in sealed vials anaerobically at $30\,°C$ under an argon atmosphere (Air Liquide Deutschland GmbH, Düsseldorf, Germany) in minimal medium containing 30 mM ethylene glycol and 120 mM $KNO_3$. Precultures were grown for 24 h before $OD_{600}$ was measured. *P. denitrificans* main cultures were inoculated to an $OD_{600}$ of 0.05, and $OD_{600}$ was monitored for 7 days. Growth was monitored by extracting $150\,\mu L$ of culture with an argon-flushed needle, and $OD_{600}$ was measured in a 96-well plate (Sarstedt AG & Co. KG, Nümbrecht, Germany) using a Tecan Infinite M Nano+ (Tecan, Männedorf, Switzerland). The growth of four independent cultures was monitored.

### Whole-cell shotgun proteomics

To acquire the proteome of *P. denitrificans* growing on different carbon sources, 30 mL cultures were grown to mid-exponential phase ($OD_{600} \sim 0.4$) in minimal medium supplemented with 60 mM glyoxylate or 60 mM ethylene glycol. Four replicate cultures were grown for each carbon source. Main cultures were inoculated from precultures grown in the same medium in a 1:1000 dilution. Cultures were harvested by centrifugation at $4000 \times g$ and $4\,°C$ for 15 min. Supernatant was discarded, and pellets were washed in 40 mL phosphate-buffered saline (PBS; 137 mM NaCl, 2.7 mM KCl, 10 mM $Na_2HPO_4$, 1.8 mM $KH_2PO_4$, pH 7.4). After washing, cell pellets were resuspended in 1 mL PBS, transferred into Eppendorf tubes, and centrifuged one more time as described above. Cell pellets in Eppendorf tubes were snap-frozen in liquid nitrogen and stored at $-80\,°C$ until they were used for the preparation of samples for LC-MS analysis and label-free quantification.

For protein extraction, bacterial cell pellets were resuspended in 4% sodium dodecyl sulfate (SDS) and lysed by heating ($95\,°C$, 15 min) and sonication (amplitude: 30%, 2 min; Hielscher Ultrasonics GmbH, Teltow, Germany). Reduction was performed for 15 min at $90\,°C$ in the presence of 5 mM tris(2-carboxyethyl)phosphine followed by alkylation using 10 mM iodoacetamide at $25\,°C$ for 30 min. The protein concentration in each sample was determined using the BCA protein assay kit (Thermo Fisher Scientific, Waltham, USA) following the manufacturer's instructions. Protein cleanup and tryptic digest were performed using the SP3 protocol[79] with minor modifications regarding protein digestion temperature and solid phase extraction of peptides. SP3 beads were obtained from GE Healthcare (Chicago, USA). $1\,\mu g$ trypsin (Promega, Fitchburg, USA) was used to digest $50\,\mu g$ of total solubilized protein from each sample. Tryptic digest was performed overnight at $30\,°C$. Subsequently, all protein digests were desalted using C18 microspin columns (Harvard Apparatus, Holliston, USA) according to the manufacturer's instructions.

LC-MS/MS analysis of protein digests was performed on a Q-Exactive Plus mass spectrometer connected to an electrospray ion source (Thermo Fisher Scientific, Waltham, USA). Peptide separation was carried out using an Ultimate 3000 nanoLC-system (Thermo Fisher Scientific, Waltham, USA), equipped with an in-house packed C18 resin column (Magic C18 AQ $2.4\,\mu m$; Dr. Maisch, Ammerbuch-Entringen, Germany). The peptides were first loaded onto a C18 pre-column (preconcentration set-up) and then eluted in backflush mode with a gradient from 94% solvent A (99.85% dd$H_2O$, 0.15% formic acid) and 6% solvent B (99.85% acetonitrile, 0.15% formic acid) to 25% solvent B over 87 min, continued with 25% to 35% of solvent B for an additional 33 min. The flow rate was set to 300 nL/min. The data acquisition mode for the initial LFQ study was set to obtain one high-resolution MS scan at a resolution of 60,000 ($m/z$ 200) with scanning range from 375 to 1500 $m/z$, followed by MS/MS scans of the 10 most intense ions. To increase the efficiency of MS/MS shots, the charged state screening modus was adjusted to exclude unassigned and singly charged ions. The dynamic exclusion duration was set to 30 s. The ion accumulation time was set to 50 ms (both MS and MS/MS). The automatic gain control (AGC) was set to $3 \times 10^6$ for MS survey scans and $1 \times 10^5$ for MS/MS scans. Label-free quantification was performed using Progenesis QI (version 2.0). MS raw files were imported into Progenesis, and the output data (MS/MS spectra) were exported in mgf format. MS/MS spectra were then searched using MASCOT (version 2.5) against a database of the predicted proteome from *P. denitrificans* downloaded from the UniProt database (www.uniprot.org; download date 01/26/2017), containing 386 common contaminant/background proteins that were manually added. The following search parameters were used: full tryptic specificity required (cleavage after lysine or arginine residues); two missed cleavages allowed; carbamidomethylation (C) set as a fixed modification; and oxidation (M) set as a variable modification. The mass tolerance was set to 10 ppm for precursor ions and 0.02 Da for fragment ions for high-energy collision dissociation (HCD). Results from the database search were imported back to Progenesis, mapping peptide identifications to MS1 features. The peak heights of all MS1 features annotated with the same peptide sequence were summed, and protein abundance was calculated per LC-MS run. Next, the data obtained from Progenesis were evaluated using the SafeQuant R-package version 2.2.2[80].

### Computational analysis of *etg* gene cluster variants

Homologs of EtgR/A/B/C were identified in other bacterial genomes via searching for the PFAM identifiers PF00171, PF00107, PF05610, and PF02954 in AnnoTree[81] with a cutoff E value of 1e-10. The lists of identified bacterial strains and homologous proteins are provided in Supplementary Data 1 and 2. The phylogenetic tree shown in Fig. 7b was downloaded from AnnoTree and annotated in Adobe Illustrator 2021. To generate Fig. 7c, sequences of selected *etg* gene clusters (given in Supplementary Data 3 and 4) were downloaded from the NCBI Nucleotide database (https://www.ncbi.nlm.nih.gov/nuccore), followed by alignment and visualization using CAGECAT/clinker[82,83] (https://cagecat.bioinformatics.nl/tools/clinker) and annotation in Adobe Illustrator 2021.

### Computational analysis of *etg*C homologs

Homologs of EtgC were identified in other bacterial genomes via BLAST[84] searching and via searching for the PFAM identifier PF05610 in AnnoTree[81] with a cutoff E value of 1e-10. To generate the phylogenetic tree shown in Supplementary Fig. 10a, sequences of EtgC homologs and related proteins (given in Supplementary Data 5) were downloaded from the NCBI Protein database (https://www.ncbi.nlm.nih.gov/protein/) and aligned using MUSCLE[85]. A maximum likelihood phylogenetic tree of the aligned sequences was calculated with MEGA X[86] using the Le-Gascuel model[87] with 100 bootstraps. The resulting tree was visualized using iTOL[88].

The alignment of amino acid sequences shown in Supplementary Fig. 10b was generated using MUSCLE and colored with Jalview[89]. Sequences used for the alignment are given in Supplementary Data 6. The superimposition of protein structures shown in Supplementary Fig. 10c was generated using Foldseek[90].

## Visualization and statistical analysis

Data were evaluated and visualized using GraphPad Prism 8.1.1., and results were compared using an unpaired *t*-test with Welch's correction in GraphPad Prism 8.1.1.

## Reporting summary

Further information on research design is available in the Nature Portfolio Reporting Summary linked to this article.

## Data availability

Genome sequencing data of evolved *P. denitrificans* isolates are available in the NCBI SRA database under accession PRJNA1126100. Mass spectrometry proteomics data are available at ProteomeXchange under accession PXD060720. Cryo-EM maps for EtgA and EtgB are deposited in the EMDB under codes EMD-50550 and EMD-50545. Atomic coordinates of EtgA and EtgB are deposited in the PDB under 9FM9 and 9FLZ. Source data are provided with this paper.

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

## Acknowledgements
We thank the staff members at the Netherlands Center for Electron Nanoscopy (NeCEN) for their help with data collection. This study was funded by Leiden University, a stipend of the Chinese Scholarship Council (grant CSC202207720003 to M.R.), and the German Research Foundation (DFG grant 446841743 to J.G.R.). T.G., J.G.R., and H.A. are grateful for the generous support of the Max Planck Society. M.H.L. was supported by a Bontius Stichting grant (BS182). Access to NeCEN was supported by the Netherlands Electron Microscopy Infrastructure (NEMI), project 184.034.014 of the National Roadmap for Large-Scale Research Infrastructure of the Dutch Research Council (NWO).

## Author contributions
M.R.: data acquisition, data analysis, figure design, writing. D.L.: data acquisition, data analysis. H.A.: data acquisition, data analysis. W.E.M.N.: data acquisition, data analysis, figure design. E.H.A.: data acquisition, data analysis. T.G.: data acquisition, data analysis. J.H.d.W.: data analysis, supervision. J.G.R.: data analysis, supervision, funding. M.H.L.: data analysis, figure design, supervision, funding. L.S.v.B.: conceptualization, data acquisition, data analysis, figure design, writing, supervision, funding.

## Competing interests
The authors declare no competing interests.
