## [Peer Review file · Nature Communications]

NAD-dependent dehydrogenases enable efficient growth of *Paracoccus denitrificans* on the PET monomer ethylene glycol

Corresponding Author: Dr Lennart Schada von Borzyskowski

Version 0:

Reviewer comments:

Reviewer #1

(Remarks to the Author)

The manuscript by Ren et al. reports the identification of promiscuous NAD-dependent dehydrogenases that promote bacterial growth on ethylene glycol. The authors employed a comprehensive set of methods to characterize key proteins (EtgB, EtgA, and EtgR) and the gene cluster they belong to. The biological roles of these proteins are well-supported by robust experimental evidence. Collectively, this work highlights a previously underappreciated role of NAD-dependent dehydrogenases in the microbial assimilation of ethylene glycol in *Paracoccus denitrificans*.

1. Although the cryo-EM structures of EtgA and EtgB do not reveal any unexpected structural features, the structure determination appears to be reliable, with resolutions of 3.0 Å and 3.1 Å and real-space correlation coefficients of 78% and 79% for the fitted models, respectively. However, the density for the fourth monomer of EtgA is incomplete. The authors should have addressed the missing density and provided a rationale or hypothesis for this observation. Overall, while the cryo-EM structures are sound, their contribution to the study is somewhat limited.

2. The authors claim that etgC encodes an iron-sulfur protein of unknown function. However, the assertion based on color alone is insufficient. They should support this claim with additional data, such as sequence comparisons, predicted structural features, and experimental validation. Techniques like ICP-MS, AAS, or conventional colorimetric assays for iron could provide stronger evidence.

3. Figure 2b: The comparison of the growth rate of *P. denitrificans* on 60 mM glycolate, 60 mM glyoxylate, and 400 mM ethylene glycol raises questions. Why was 60 mM chosen for glycolate and glyoxylate? Is this the optimal concentration for *P. denitrificans* growth? Clarification is needed.

4. Page 7, Line 123: In the proteome analysis of *P. denitrificans* growing on either ethylene glycol or glyoxylate, the concentration of EG used was 60 mM. Why was this concentration chosen over the presumably optimal 400 mM?

5. Page 11, Line 166: Supplementary Table 1 should include the kinetic parameters of EtgB for completeness.

6. Page 11, Line 171: The manuscript lacks information on the oxidation and reduction activities of EtgB at pH levels below 6.0 and above 10.0. These data would provide a more complete picture.

7. Page 11, Line 173: How long does the enzyme maintain its activity at temperatures above 50°C? This information would be valuable for understanding the enzyme's thermal stability.

8. Page 12, Line 199: It would be beneficial to conduct a Y464G mutation, as it may provide further insights into the mechanism of activity.

9. The figures, particularly those with significant blank space, could be improved by adding simple labels. For instance, in Figure 2, labeling the three curves directly would make the figure more intuitive. Additionally, bar graphs might offer a clearer visualization of the data, as the current mean (black line) is not clearly visible. The authors should consider revisiting all

relevant figures to enhance their readability and accessibility.

10. Page 18, Line 278: The expression "we hypothesized that growth on the latter compound is limited by the activity of EtgB and EtgA" is vague. It should be clarified.

11. Supplementary Figure 6a-e: The statistical significance of the comparisons should be indicated. Additionally, the figure could be better organized by directly comparing the activities of EtgB and EtgA in different evolved strains grown on succinate or EG.

12. The manuscript used varying concentrations of EG in different assays (400 mM, 200 mM, 60 mM). The rationale behind the selection of these concentrations should be explained to avoid confusion.

13. Supplementary Figure 7: Data from the wild type strain should be included for comparison.

14. Page 20, Line 309: The phrase "were consistently 2 to 3 times higher than those in the WT" is unclear. The specific kinetic parameters used for this comparison should be mentioned.

15. Page 21, Figure 7: The meaning of the grey arrows is not explained. A legend or brief explanation should be provided.

16. Accession codes for the proteins studied should be listed.

Reviewer #2

(Remarks to the Author)

The authors present a very interesting paper on the NAD-dependent utilization of EG by cells. This is a new/underappreciated pathway and the authors have done a nice job characterizing/identifying this cluster, performing kinetics and crystallography analysis and proposing this pathway as essential through adaptive evolution experiments. There are a few topics of further refinement here for the manuscript:

Figure 2 and its conclusions need additional statistical justification. The authors state the optimal concentration of 400 μ M for growth, but there is not statistical treatment to see if this is indeed a single point optimum or if there is a range of optimal concentrations from roughly 200-600 μ M. Moreover, the characterization of growth in Fig 2B is showing an increased lag phase, but all 3 carbon sources arrive at the same final carrying capacity. This should be discussed more. Is there a toxicity effect or something else?

The authors use p-values in the paper in figures such as Fig 6, but do not state what type of statistical test was used to calculate these. Likewise, throughout it indicates $n=3$, but are these biological or technical?

For the resequencing effort of the evolved strains, are the point mutations seen only the ones in the table or the ones that the authors investigated further? The total number of SNPs would be important to see as well.

Reviewer #3

(Remarks to the Author)

The submitted manuscript describes the discovery and characterisation of the *etg* gene cluster responsible for the ability of the soil bacterium *Paracoccus denitrificans* to grow on ethylene glycol. Using comparative proteomic analysis of *P. denitrificans* growing on either ethylene glycol or glyoxylate, they identify a gene cluster comprised of a NAD-dependent aldehyde dehydrogenase *etgA*, a NAD-dependent alcohol dehydrogenase *etgB*, and a third gene encoding for the protein *etgC* with unknown function.

The authors also characterise the role of both NAD-dependent enzymes during the metabolic assimilation of ethylene glycol as well as the role of a gene adjacent to the *etgABC* cluster, called *etgR*. This gene encodes for the transcriptional regulator *EtgR* responsible for activating the expression of the *etg* genes in the presence of ethylene glycol.

This work will, therefore, interest a wide audience working in the active field of plastic re- and up-cycling and therefore should be considered for publication in *Nature Communications* after addressing the following major points of concern:

- The authors claim that this *etg* gene cluster is widely distributed among proteobacteria but only includes alpha and gammaproteobacteria, and only three of these have the alcohol dehydrogenase (also not many gamma-). They should do more bioinformatic analysis.

- It is claimed several times in the text that *EtgB* has increased activity. Still, experiments showed on this work were done with cell-free extract, therefore, protein concentration isn't normalised, and it could be overexpression of the enzyme rather than higher enzyme activity per se (specific activity could not be calculated, so it is not right to use U/mg) (Line 105, line 344)

- Line 310: More active enzyme or more expression of the enzymes? It would be helpful to add SDS-PAGE gel.

- Line 115: Growth was “considerably slower”, but EG concentration was 6.6x more concentrated than the others. Why different concentrations? Is it directly comparable?
 - Line 117 (Figure 2): a) Why use a specific growth rate without doubling time? This isn't directly comparable with the example of FucO in the introduction. It should be consistent either way. It should show error bars, not dots, and include bars or boxes. b) No measure of error on graph (but 6 replicates) – should have linear regression or show error bars.
 - It should say how the proteome was analysed; this is the most important part of the paper (Lines 123 and 145).
 - Does heterologous expression in *E. coli* confer growth on EG? (Line 158)
 - Line 188: The mutations weren't very extensive – only two were tested, and this mutation didn't give too much information. As we understand, the mutations were done with the hypothesis that the changes of Thr44Ser and His44Asn would improve the conversion of ethylene glycol and other diols. However, the results show a decrease in the activity towards ethylene glycol. So why is this happening?
- So, consider that this part should be removed; otherwise, it needs more experiments.
- Line 197: Do you have any data to show structural similarity? How does EtgA compare with other aldehyde dehydrogenases by pairwise structural alignment?
 - Line 228: This section needs more characterisation work to be included. Did they get a structure for this (even AlphaFold)? If so, how does the structure compare with known enzymes? Are there any iron-sulphur clusters containing enzymes with known function in proximity to EtgB homologues?
 - Line 320: Do you have any evidence to show this?
 - Line 321: Only two *Pseudomonas* alignments are shown. Two isn't “several”
 - Line 328 (Figure 7): How did you select these strains for alignment? A more comprehensive range of alignments would be better, including any alignments in *E. coli*, as you have made several references to *E. coli* in the paper, as well as bacteria from other classes
 - Line 357: If it is likely that EtgB can be engineered to optimise activity, why claim it was already well suited to the reaction (line 188)?
 - Line 366: If this is a promising enzyme for similar in vivo engineering projects, what was the result of expressing it in *E. coli*? Did it enable growth on EG?

Minor Comments:

- There is no great emphasis on enzyme promiscuity, considering this is the title.
- Line 15: It should be polyethylene terephthalate (PET), not the other way around.
- Line 34: 6.3 billion
- Line 41: More examples of PETases, there is just one reference.
- Line 75: Include the release of CO₂ in Figure 1.
- Line 91: Could you include the doubling time of *P. Putida* on EG?
- Line 97: Confusing word order, they say “both” when there are three things in that clause.
- Line 143 (Figure 3b): The graph is out of frame (bars don't line up, and the top line isn't matched up).
- Lines 151 & 152: The final sentence is not appropriate for a figure legend. Observations should be in the results.
- Line 158: “expressed”, not “produced”.
- Line 172: At what temperature does the enzyme plateau (S3a)? And where is the data that activity is lost at 50 °C?
- Line 189: By how much is catalytic efficiency improved for propylene glycol?
- Line 205 (Table 1): Km should be in the same unit as supplementary table 1 for easy comparison.
- Line 206 (Figure 4): a and b should be combined, as there is a comparison between them. (same as Figure 5)
- Line 258: Again, observations shouldn't be included in the figure legend.
- Line 262: If the experiments were repeated three times, should include data. What is the standard deviation?
- Line 291 (Table 2): All tables in the document should be in the same format
- Line 324: “may” play an underappreciated role, “likely” is too much of a claim
- Line 424: What RPM for shaking?
- Line 486: What version of AlphaFold did you use?
- Line 528: How many times was it centrifuged?
- Line 532: What were the sonication conditions?
- Line 548: Is this 99.85% H₂O?

Reviewer #4

(Remarks to the Author)

The manuscript by Ren et al. describes the bacterial catabolism of ethylene glycol, a small alcohol derived from PET. The authors used a combination of approaches aimed at identifying EtgB and EtgA as enzymes responsible for the NAD⁺-dependent dehydrogenation of ethylene glycol and glycolaldehyde, respectively, in *Paracoccus denitrificans*. The authors further present structures of the two enzymes and investigate the transcriptional regulation of the *etg* genes. The authors also investigated the phylogenetic distribution of the pathway and used adaptive laboratory evolution to improve the strain's growth on ethylene glycol. The manuscript is clearly written and the data are clearly presented. Unfortunately, important experiments and controls are missing. Moreover, some of the data are not properly interpreted. Most importantly, the presented data do not support the conclusion that EtgB is physiologically relevant to ethylene glycol catabolism. For example, the presented data do not rule out that only one of EtgA or EtgB is involved in ethylene glycol catabolism, nor do they rule out that the PQQ-dependent methanol dehydrogenase is involved. These possibilities need to be investigated in light of the very low specificity of EtgB for ethylene glycol, the well-established role of PQQ-dependent enzymes in ethylene glycol dehydrogenation, and the known ability of glycolaldehyde dehydrogenases in improving growth of strains on ethylene glycol.

Specific comments:

1. In the Abstract, “unknown” should be “uncharacterized” or “unrecognized” (l. 21).
2. In discussing the potential of biocatalysts for upcycling plastics, the author should cite Sullivan et al. (2022; 10.1126/science.abo4626) (l. 47). This study provides proof-of-concept for the upcycling of mixed plastics waste, including PET.
3. The values and errors in Table 1 should be cited to the correct number of significant figures. Some values are cited to five significant figures.
4. Errors should be provided for the k_{cat}/K_m values (Table 1). These errors should be calculated directly from the data, not from the errors of the k_{cat} and K_m values.
5. The experimental conditions (buffer, pH, temperature) used in determining the steady-state kinetic parameters should be provided as a table note (Table 1).
6. The V_{max} values should be deleted from Table 1 – they are redundant with the k_{cat} values.
7. In comparing the ability of EtgB to act on different substrates, the authors should compare k_{cat}/K_m values, which are a measure of substrate specificity, and not K_m (ll. 163-164, l. 182).
8. It is unclear whether EtgB oxidizes ethylene glycol significantly more efficiently than other alcohol dehydrogenases because the steady-state parameters that are compared were determined under such different conditions (p. 11). Notably, those of EtgB were determined at pH 10, its optimal pH, and not at a physiologically-relevant pH value. In this respect, the substrate specificity of glycerol dehydrogenase of *Thermus thermophilus* for ethylene glycol appears to be quite similar to that of EtgB (Table S1).
9. Supplementary Figure S2 needs to be cited in the text (l. 171).
10. In discussing the structural similarity of EtgB and EtgA with other enzymes, the authors should cite RMSD values or some other objective measure (p. 12). It should also be clarified which enzymes are most structurally similar to EtgB and EtgA.
11. Characterization of gene deletion mutants must include the phenotypes of complemented strains as a control.
12. When reporting Results, past tense should be used (l. 258).
13. The role of EtgR cannot be deduced from the growth data of the mutant alone (l. 249). The “conclusion” needs to be revised.
14. Given the high substrate specificities of EtgB and EtgA for acetaldehyde, the growth of the ΔetgR mutant strains and their complements should be tested on this substrate (Fig. 6a).
15. Similarly, the promoter reporter strain should be characterized using acetaldehyde (Fig. 6b).
16. “Variants of the *etg* gene cluster are...” should be “The *etg* genes are...” (l. 314).
17. References establishing ethylene glycol as a next-generation feedstock in microbial biotechnology should be cited in the Introduction (l. 334).
18. The statement that EtgB catalyzes the conversion of ethylene glycol to glycolaldehyde with high efficiency is incorrect (l. 341). Most importantly, the substrate specificity (k_{cat}/K_m) of EtgB for ethylene glycol is very low (650 $\text{M}^{-1} \text{s}^{-1}$) and the values for ethanol and acetaldehyde are ~300-fold and ~6,000-fold higher, respectively.
19. The presented data do not support the conclusion that “increased activity of EtgB and EtgA result in faster growth of the evolved strains on ethylene glycol” (ll. 344-345). For example, the inability of the ΔetgR mutant strains to grow on ethylene glycol does not test the hypothesis that only one of *etgA* or *etgB* is involved in ethylene glycol catabolism. Similarly, the analysis of the adaptive laboratory evolution strains does not test this hypothesis. Based on what is known about ethylene glycol catabolism in *Pseudomonas* and *Rhodococcus*, it is reasonable to hypothesize that in *P. denitrificans*, the PQQ-dependent methanol dehydrogenase and EtgA catalyze the dehydrogenation of ethylene glycol and glycolaldehyde, respectively. Glycolaldehyde is cytotoxic, and its transformation is rate-limiting in many strains. This hypothesis is consistent with the adaptive laboratory evolution data in *P. denitrificans* (i.e., increased production of EtgA would decrease the intracellular concentration of glycolaldehyde).

Reviewer comments:

Reviewer #1

(Remarks to the Author)

My concerns have been addressed. Thanks authors' efforts.

Reviewer #2

(Remarks to the Author)

The authors have nicely revised their manuscript and addressed all the concerns that I had raised in the previous round of review.

Reviewer #3

(Remarks to the Author)

I am pleased to see the authors have taken the time to address all the comments in our original rebuttal. The study and manuscript are now much more complete, coherent and impactful. I therefore recommend publication after the authors address the following minor remaining edits:

- Error bars should be included in all growth curves. This can be made less distracting by using a light coloration relative to the plotted data and will be more informative to illustrate any significant differences to the reader.
- the authors mention that the majority of K_m values in Table 1 are in the low μM range, but 11 of the 20 are in the mM range. I think this should be changed to mM for consistency with Supplementary Table 1.
- Line 52: as a source of carbon
- Line 101: which then enters central carbon metabolism...
- Line 186: with a k_{cat}/K_m value that is 300-fold lower.

Reviewer #4

(Remarks to the Author)

The revised manuscript contains a considerable amount of new data and analyses that address my comments on the original submission. This more complete study significantly advances our understanding of the bacterial catabolism of ethylene glycol.

REVIEWER COMMENTS

Reviewer #1 (Remarks to the Author):

The manuscript by Ren et al. reports the identification of promiscuous NAD-dependent dehydrogenases that promote bacterial growth on ethylene glycol. The authors employed a comprehensive set of methods to characterize key proteins (EtgB, EtgA, and EtgR) and the gene cluster they belong to. The biological roles of these proteins are well-supported by robust experimental evidence. Collectively, this work highlights a previously underappreciated role of NAD-dependent dehydrogenases in the microbial assimilation of ethylene glycol in *Paracoccus denitrificans*.

Thank you for the positive assessment of our study.

1. Although the cryo-EM structures of EtgA and EtgB do not reveal any unexpected structural features, the structure determination appears to be reliable, with resolutions of 3.0 Å and 3.1 Å and real-space correlation coefficients of 78% and 79% for the fitted models, respectively. However, the density for the fourth monomer of EtgA is incomplete. The authors should have addressed the missing density and provided a rationale or hypothesis for this observation. Overall, while the cryo-EM structures are sound, their contribution to the study is somewhat limited.

We have included an analysis of the inter-subunit surfaces of EtgA in our revised manuscript, which indicates that the EtgA tetramer consists of a dimer-of-dimers, in which the inter-dimer contact is weak. This provides a plausible explanation for the missing density, as the weak contact between the two dimers would result in movement between the two dimers, which in turn would result in a poor density. This analysis is shown in Supplementary Figure 7.

2. The authors claim that *etgC* encodes an iron-sulfur protein of unknown function. However, the assertion based on color alone is insufficient. They should support this claim with additional data, such as sequence comparisons, predicted structural features, and experimental validation. Techniques like ICP-MS, AAS, or conventional colorimetric assays for iron could provide stronger evidence.

To address this comment, we have performed additional computational and experimental work on EtgC. Phylogenetic and structural analysis revealed that EtgC is related to iron-sulfur cluster assembly proteins and ferredoxins, and sequence analysis demonstrated the presence of four conserved cysteines that often serve to coordinate an iron-sulfur cluster (Supplementary Figure 8). However, ICP-OES analysis of purified EtgC did not detect any iron (Supplementary Table 4).

Therefore, we conclude that EtgC is either not an iron-sulfur cluster protein, or the iron atom(s) were not correctly inserted into the heterologously produced protein. Since we also show that deletion of *etgC* does not have a strong impact on growth of *P. denitrificans* on ethylene glycol (Supplementary Figure 9), we conclude that the further investigation of this protein of unknown function is beyond the scope of the current manuscript.

3. Figure 2b: The comparison of the growth rate of *P. denitrificans* on 60 mM glycolate, 60 mM glyoxylate, and 400 mM ethylene glycol raises questions. Why was 60 mM chosen for glycolate and glyoxylate? Is this the optimal concentration for *P. denitrificans* growth? Clarification is needed.

In Figure 2b, we aimed to visually compare the relatively fast growth of *P. denitrificans* on 60 mM glycolate or 60 mM glyoxylate (which was established in previous work from us; <https://doi.org/10.1038/s41586-019-1748-4>) with the slower growth on ethylene glycol. To this end, we chose to show the growth curve of *P. denitrificans* on 400 mM ethylene glycol, since it allows faster

growth than other concentrations of ethylene glycol (see Figure 2a), which is still considerably slower than growth on glyoxylate or glyoxylate.

4. Page 7, Line 123: In the proteome analysis of *P. denitrificans* growing on either ethylene glycol or glyoxylate, the concentration of EG used was 60 mM. Why was this concentration chosen over the presumably optimal 400 mM?

For the proteomics experiment, we deemed it more relevant to have equal concentrations of the carbon sources (60 mM) instead of fastest growth on ethylene glycol. Therefore, we chose 60 mM ethylene glycol instead of 400 mM ethylene glycol to provide the same concentration of the carbon source for both glyoxylate and ethylene glycol.

5. Page 11, Line 166: Supplementary Table 1 should include the kinetic parameters of EtgB for completeness.

We added the kinetic parameters of EtgB (now determined at two different pH values) to Supplementary Table 1.

6. Page 11, Line 171: The manuscript lacks information on the oxidation and reduction activities of EtgB at pH levels below 6.0 and above 10.0. These data would provide a more complete picture.

We performed additional experiments to determine oxidation and reduction activities of EtgB at the pH values 3, 4, 5, and 11. We added these new data points to Supplementary Figure 3.

7. Page 11, Line 173: How long does the enzyme maintain its activity at temperatures above 50°C? This information would be valuable for understanding the enzyme's thermal stability.

We replaced the imprecise statement about the activity of EtgB above 50 °C with new experimental data on the thermal stability of this enzyme (and also on the stability of EtgA and EtgC; see Supplementary Figure 6f).

8. Page 12, Line 199: It would be beneficial to conduct a Y464G mutation, as it may provide further insights into the mechanism of activity.

We introduced the Y464G mutation in EtgA and attempted to characterize enzyme activity of the resulting mutant enzyme. The Y464G mutant did not show activity with both acetaldehyde and glycolaldehyde, which further strengthens the idea that this residue is important for the conversion of smaller aldehyde substrates (see L. 233-235).

9. The figures, particularly those with significant blank space, could be improved by adding simple labels. For instance, in Figure 2, labeling the three curves directly would make the figure more intuitive. Additionally, bar graphs might offer a clearer visualization of the data, as the current mean (black line) is not clearly visible. The authors should consider revisiting all relevant figures to enhance their readability and accessibility.

We revisited our figures and improved them, for example by the addition of labels or the clear visualization of the mean value, where applicable. Furthermore, the values of each sample/data point can be found in the relevant Source Data files, which we are providing together with the revised manuscript.

We are not using any bar graphs in this study, since this is strongly discouraged by *Nature Communications* ("Please replace your bar graphs with plots that feature information about the distribution of the underlying data. All data points should be shown for plots with a sample size less than 10.").

10. Page 18, Line 278: The expression "we hypothesized that growth on the latter compound is limited by the activity of EtgB and EtgA" is vague. It should be clarified.

We rephrased this sentence to clarify the intended statement.

11. Supplementary Figure 6a-e: The statistical significance of the comparisons should be indicated. Additionally, the figure could be better organized by directly comparing the activities of EtgB and EtgA in different evolved strains grown on succinate or EG.

We improved this figure (now Supplementary Figure 11) as suggested.

12. The manuscript used varying concentrations of EG in different assays (400 mM, 200 mM, 60 mM). The rationale behind the selection of these concentrations should be explained to avoid confusion.

We have previously explained our choices for the ethylene glycol concentration in Figure 2b and the proteomics experiments in points 3 and 4 of these replies (see above). For the adaptive laboratory evolution experiment we chose 200 mM ethylene glycol to achieve a balance between fast growth and limited carbon availability, since we reasoned that this might result in a suitable selective pressure to elicit mutations that improve growth of *P. denitrificans* on ethylene glycol.

13. Supplementary Figure 7: Data from the wild type strain should be included for comparison.

For comparison, we also added the relevant data from the WT strain to this figure (now Supplementary Figure 12).

14. Page 20, Line 309: The phrase "were consistently 2 to 3 times higher than those in the WT" is unclear. The specific kinetic parameters used for this comparison should be mentioned.

We rephrased this sentence to clarify the statement about increased enzyme activities in the evolved strains. However, we do not think that eight numerical values (two enzyme activities measured in four different strains) should be included in this sentence. This would make the text unnecessarily hard to follow for the reader. The exact values of the measured enzyme activities can be found in the relevant Source Data files, which we are providing together with the revised manuscript.

15. Page 21, Figure 7: The meaning of the grey arrows is not explained. A legend or brief explanation should be provided.

We are providing an improved version of Figure 7 and its legend in the revised version of our manuscript.

16. Accession codes for the proteins studied should be listed.

We have added the Uniprot accession codes for EtgA/B/C/R in the revised version of the manuscript (see L. 135-137, L. 292).

Reviewer #2 (Remarks to the Author):

The authors present a very interesting paper on the NAD-dependent utilization of EG by cells. This is a new/underappreciated pathway and the authors have done a nice job characterizing/identifying this cluster, performing kinetics and crystallography analysis and proposing this pathway as essential through adaptive evolution experiments. There are a few topics of further refinement here for the manuscript:

Figure 2 and its conclusions need additional statistical justification. The authors state the optimal concentration of 400 mM for growth, but there is not statistical treatment to see if this is indeed a single point optimum or if there is a range of optimal concentrations from roughly 200-600 mM.

Thank you for pointing this out. We have performed statistical tests to compare the growth rates of *P. denitrificans* on different concentrations of ethylene glycol. This shows that there are significant differences between the growth rates on all concentration steps of ethylene glycol (i.e., a significant difference between the growth rate on 50 and 100 mM ethylene glycol, and between 100 and 200 mM, and between 200 and 400 mM, and between 400 and 600 mM, and between 600 and 900 mM, and between 900 and 1200 mM). We have denoted this in the revised version of Figure 2 and its legend.

Moreover, the characterization of growth in Fig 2B is showing an increased lag phase, but all 3 carbon sources arrive at the same final carrying capacity. This should be discussed more. Is there a toxicity effect or something else?

We have extended the discussion of these results (see L. 117-119). In brief, an increased lag phase during growth on ethylene glycol (when compared to growth on glycolate or glyoxylate) was previously observed for *Pseudomonas putida* and ascribed to the required induction of additional enzymes (<https://doi.org/10.1128/AEM.02062-12>), which is consistent with our findings for *P. denitrificans*.

The authors use p-values in the paper in figures such as Fig 6, but do not state what type of statistical test was used to calculate these. Likewise, throughout it indicates n=3, but are these biological or technical?

We regret that we did not add this information about statistical tests. In the revised version of the manuscript, we included all relevant information about data visualization and statistical tests in the Methods section, and details about replicates and independent experiments can be found in the figure legends and in the separate 'Reporting summary' document.

For the resequencing effort of the evolved strains, are the point mutations seen only the ones in the table or the ones that the authors investigated further? The total number of SNPs would be important to see as well.

Table 2 already shows all point mutations that we observed in the genome resequencing data of the evolved strains.

Reviewer #3 (Remarks to the Author):

The submitted manuscript describes the discovery and characterisation of the *etg* gene cluster responsible for the ability of the soil bacterium *Paracoccus denitrificans* to grow on ethylene glycol. Using comparative proteomic analysis of *P. denitrificans* growing on either ethylene glycol or glyoxylate, they identify a gene cluster comprised of a NAD-dependent aldehyde dehydrogenase *etgA*, a NAD-dependent alcohol dehydrogenase *etgB*, and a third gene encoding for the protein *EtgC* with unknown function.

The authors also characterise the role of both NAD-dependent enzymes during the metabolic assimilation of ethylene glycol as well as the role of a gene adjacent to the *etgABC* cluster, called *etgR*. This gene encodes for the transcriptional regulator *EtgR* responsible for activating the expression of the *etg* genes in the presence of ethylene glycol.

This work will, therefore, interest a wide audience working in the active field of plastic re- and up-cycling and therefore should be considered for publication in *Nature Communications* after addressing the following major points of concern:

- The authors claim that this *etg* gene cluster is widely distributed among proteobacteria but only includes alpha and gammaproteobacteria, and only three of these have the alcohol dehydrogenase (also not many gamma-). They should do more bioinformatic analysis.

We have extended our bioinformatic analyses considerably in the revised version of the manuscript (see Figure 7 and accompanying text). We now show that homologs of the *etgRABC* genes are present in 3793 strains, mainly among Alpha- and Gammaproteobacteria as well as Bacilli, Actinomycetes, and Bacteroidia. It is not possible to visualize the *etg* genes in several thousand bacterial strains; therefore, our analysis highlights that there are diverse variants of the gene cluster, including *etgABC* or only *etgAC* together with *etgR* or a gene encoding for a different transcriptional regulator.

- It is claimed several times in the text that *EtgB* has increased activity. Still, experiments showed on this work were done with cell-free extract, therefore, protein concentration isn't normalised, and it could be overexpression of the enzyme rather than higher enzyme activity per se (specific activity could not be calculated, so it is not right to use U/mg) (Line 105, line 344)

We believe that this comment might be due to a misunderstanding, caused by imprecise phrasing on our part. We have indeed observed increased activity of *EtgB* and *EtgA* in cell-free extracts of evolved strains grown on ethylene glycol (measured in U per mg of total protein; see revised Supplementary Figure 11). This is linked to increased production of *EtgB* and *EtgA* (as shown by our proteomics data; see Supplementary Figure 1). However, we do not claim that the enzyme *EtgB* itself has increased activity in the evolved strains (since there are no mutations found in the *etgB* gene in any of the evolved strains). We state that the increased activity in the cell-free extracts is solely due to increased enzyme production, and have now clarified this (see L. 354-356, L. 364-369).

- Line 310: More active enzyme or more expression of the enzymes? It would be helpful to add SDS-PAGE gel.

As stated in the previous reply, we think that the increased activity of *EtgB* and *EtgA* in the cell-free extracts is solely due to increased enzyme production. This is supported by our proteomics data (shown in Supplementary Figure 1). We did not add an SDS-PAGE gel, since the information on protein production levels gained from mass spectrometry-based proteomics data can be considered more reliable than protein band size on a gel.

- Line 115: Growth was “considerably slower”, but EG concentration was 6.6x more concentrated than the others. Why different concentrations? Is it directly comparable?

As shown in Figure 2a, we also tested growth of *P. denitrificans* on other concentrations of ethylene glycol, both lower and higher than 400 mM. In all cases, the average growth rate was significantly lower than the growth rate on 400 mM ethylene glycol. In Figure 2b, we chose to show the growth curve on 400 mM ethylene glycol to visualize that even the best growth rate on ethylene glycol is considerably slower than growth on other two-carbon substrates (glycolate and glyoxylate). The standard concentration of 60 mM for glycolate and glyoxylate was established in previous work from us (see <https://doi.org/10.1038/s41586-019-1748-4>).

- Line 117 (Figure 2): a) Why use a specific growth rate without doubling time? This isn't directly comparable with the example of FucO in the introduction. It should be consistent either way. It should show error bars, not dots, and include bars or boxes. b) No measure of error on graph (but 6 replicates) – should have linear regression or show error bars.

We have made this more consistent by also giving the growth rate for FucO-mediated growth on ethylene glycol in the introduction (L. 68/69), calculated via the equation

Growth rate = $\text{LN}(2)/\text{doubling time}$

We have also improved Figure 2 by making the mean values and error bars (which were already present) better visible.

We are not using any bar graphs in this study, since this is strongly discouraged by *Nature Communications* (“Please replace your bar graphs with plots that feature information about the distribution of the underlying data. All data points should be shown for plots with a sample size less than 10.”). Therefore, we are showing all data points as dots in Figure 2a.

In Figure 2b, we have chosen to show representative growth curves from among the six replicates, with errors < 5%. There are ca. 1,000 measured data points that make up the growth curves shown here; if we would show error bars for all of them, it would make the figure hard to interpret for readers, since a) the tiny error bars would barely be visible and b) the curves and their error bars would partly overlap, which hides relevant information.

- It should say how the proteome was analysed; this is the most important part of the paper (Lines 123 and 145).

The proteome was analyzed via mass spectrometry-based whole cell shotgun proteomics, which is now specified in the main text (L. 128). All experimental details for this method are given in the ‘Materials and methods’ section.

- Does heterologous expression in *E. coli* confer growth on EG? (Line 158)

In the revised version of our manuscript, we present additional data that demonstrate growth of *E. coli* with heterologously expressed *etgB* on ethylene glycol (see Supplementary Figure 5).

- Line 188: The mutations weren't very extensive – only two were tested, and this mutation didn't give too much information. As we understand, the mutations were done with the hypothesis that the changes of Thr44Ser and His44Asn would improve the conversion of ethylene glycol and other diols. However, the results show a decrease in the activity towards ethylene glycol. So why is this happening?

As shown by our data, the activity of the three tested mutants towards ethylene glycol is decreased, but the activity towards propylene glycol is increased. We hypothesize that these mutations have

opened up the active site to better accommodate the bulky molecule propylene glycol, while ethylene glycol might not be optimally positioned in the active site of these mutant variants of EtgB. We now clarify this in the revised manuscript (L. 214-218).

So, consider that this part should be removed; otherwise, it needs more experiments.

We have chosen not to remove this part, also due to the fact that reviewer 1 requested to test a mutant variant of EtgA as well. While these mutation experiments are limited in scope, they offer the first insights into the active sites of EtgB and EtgA. We plan to follow this up with a more detailed analysis and comprehensive engineering efforts of the active sites; but this approach goes beyond the scope of the current study.

- Line 197: Do you have any data to show structural similarity? How does EtgA compare with other aldehyde dehydrogenases by pairwise structural alignment?

We have listed enzymes with high structural similarity to EtgA (and also enzymes with high structural similarity to EtgB and EtgC) in Supplementary Table 3 in the revised version of the manuscript.

- Line 228: This section needs more characterisation work to be included. Did they get a structure for this (even AlphaFold)? If so, how does the structure compare with known enzymes? Are there any iron-sulphur clusters containing enzymes with known function in proximity to EtgB homologues?

To address this comment, we have performed additional computational and experimental work on EtgC. Phylogenetic and structural analysis revealed that EtgC is related to iron-sulfur cluster assembly proteins and ferredoxins, and sequence analysis demonstrated the presence of four conserved cysteines that often serve to coordinate an iron-sulfur cluster (Supplementary Figure 8). However, ICP-OES analysis of purified EtgC did not detect any iron (Supplementary Table 4).

Therefore, we conclude that EtgC is either not an iron-sulfur cluster protein, or the iron atom(s) were not correctly inserted into the heterologously produced protein. Since we also show that deletion of *etgC* does not have a strong impact on growth of *P. denitrificans* on ethylene glycol (Supplementary Figure 9), we conclude that the further investigation of this protein of unknown function is beyond the scope of the current manuscript.

- Line 320: Do you have any evidence to show this?

This comment refers to the statement: 'In these cases, the role of EtgB might be taken over by alcohol dehydrogenases encoded at other locations in the genome.'

As stated in an earlier comment, our extended bioinformatic analysis shows that there are 3793 bacterial strains with homologs of the *etgRABC* genes. Many of the examples highlighted in Figure 7c only have the genes *etgA*, *etgC*, and *etgR* (or a gene encoding for a different transcriptional regulator) in close genomic proximity. Therefore, these strains do have a homolog of *etgB* which is located elsewhere in their genome, which we now also point out in the revised manuscript (L. 382).

- Line 321: Only two *Pseudomonas* alignments are shown. Two isn't "several"

We have revised this section (L. 371-385) as well as Figure 7 to better address the results of our revised bioinformatics analysis regarding the phylogenetic distribution of the *etg* genes.

- Line 328 (Figure 7): How did you select these strains for alignment? A more comprehensive range of alignments would be better, including any alignments in *E. coli*, as you have made several references to *E. coli* in the paper, as well as bacteria from other classes

As suggested, we have added additional alignments to the revised version of Figure 7 (including bacteria from seven different classes). However, we cannot add an alignment including *E. coli*, since *E. coli* does not have any homologs of the *etg* genes. It is correct that we are making references to the ethylene glycol metabolism of *E. coli* in our study; but in this model bacterium, oxidation of ethylene glycol to glycolate is mediated by the enzymes encoded by the *fucO* and *aldA* genes, as described in our study (L. 68, L. 132/133).

- Line 357: If it is likely that EtgB can be engineered to optimise activity, why claim it was already well suited to the reaction (line 188)?

We have removed the statement that EtgB is already well suited to the conversion of ethylene glycol in order to address this comment.

Furthermore, we have added a statement to the discussion (L. 421-423) regarding the relatively good growth rate of *P. denitrificans* on ethylene glycol with only NAD-dependent enzymes (since we now show that the PQQ-dependent enzyme MxaF is not involved in ethylene glycol metabolism). The key factors for this phenotype are likely high production of EtgB, and a low K_M of EtgA for glycolaldehyde.

- Line 366: If this is a promising enzyme for similar in vivo engineering projects, what was the result of expressing it in *E. coli*? Did it enable growth on EG?

In the revised version of our manuscript, we present additional data that demonstrate growth of *E. coli* with heterologously expressed *etgB* on ethylene glycol (see Supplementary Figure 5).

Minor Comments:

We have addressed all these minor comments, where appropriate. The relevant lines in the revised version of the manuscript are given below.

- There is no great emphasis on enzyme promiscuity, considering this is the title.

We changed the title of the manuscript to 'NAD-dependent dehydrogenases enable efficient growth of the soil bacterium *Paracoccus denitrificans* on the PET monomer ethylene glycol'.

- Line 15: It should be polyethylene terephthalate (PET), not the other way around.

Changed as suggested; L. 15/16.

- Line 34: 6.3 billion

Changed as suggested; L. 34.

- Line 41: More examples of PETases, there is just one reference.

We have added references for two additional PETase derivatives, Combi-PETase and HotPETase.

- Line 75: Include the release of CO₂ in Figure 1.

Release of CO₂ is not taking place in any of the reactions that are shown in Figure 1. Release of CO₂ is taking place in the glycerate pathway and the TCA cycle, but we do not intend to show these pathways and their reactions in detail in Figure 1, since it would take up the majority of space and distract from the ethylene glycol-converting reactions, which are the main focus of this figure.

- Line 91: Could you include the doubling time of *P. Putida* on EG?

We have included the growth rate of *P. putida* here; L. 92.

- Line 97: Confusing word order, they say “both” when there are three things in that clause.

We have rephrased this sentence to avoid confusion; L. 98.

- Line 143 (Figure 3b): The graph is out of frame (bars don't line up, and the top line isn't matched up).

We have revised Figure 3b to ensure that the top line is matched up better. We cannot line up the bars neatly, because there are 18 bars (separated by three lines) for the upregulated proteins and 19 bars (separated by ten lines) for the downregulated proteins.

- Lines 151 & 152: The final sentence is not appropriate for a figure legend. Observations should be in the results.

We have removed this sentence.

- Line 158: “expressed”, not “produced”.

In this study, we follow the common convention that genes are expressed, while proteins are produced. Since we are referring to proteins in this sentence, the phrasing ‘we produced both enzymes in *E. coli*’ is in our opinion correct.

- Line 172: At what temperature does the enzyme plateau (S3a)? And where is the data that activity is lost at 50 °C?

We replaced the imprecise statement about the activity of EtgB above 50 °C with new experimental data on the thermal stability of this enzyme (and also on the stability of EtgA and EtgC; see Supplementary Figure 6f).

- Line 189: By how much is catalytic efficiency improved for propylene glycol?

The catalytic efficiency for propylene glycol is improved ca. three-fold for the H47N mutant of EtgB and ca. five-fold for the T44S H47N mutant of EtgB, compared to EtgB WT. We have added this information now (L. 213/214).

- Line 205 (Table 1): Km should be in the same unit as supplementary table 1 for easy comparison.

In Supplementary Table 1, all Km values are in the millimolar range. In Table 1, the majority of Km values are in the (low) micromolar range. Therefore, we have chosen to give Km values in μM for Table 1, but in mM for Supplementary Table 1.

- Line 206 (Figure 4): a and b should be combined, as there is a comparison between them. (same as Figure 5)

Changed as suggested for Figures 4 and 5.

- Line 258: Again, observations shouldn't be included in the figure legend.

Changed as suggested.

- Line 262: If the experiments were repeated three times, should include data. What is the standard deviation?

In Figure 6b, we have chosen to show representative growth curves from among the three replicates, with errors < 5%. There are ca. 400 measured data points that make up the growth and fluorescence curves shown here; if we would show three curves in each case, or error bars for datapoints, it would

make the figure hard to interpret for readers, since a) the tiny error bars would barely be visible and b) the curves or their error bars would partly overlap, which hides relevant information.

- Line 291 (Table 2): All tables in the document should be in the same format

We adapted Table 2 so as to have the same format as Table 1.

- Line 324: “may” play an underappreciated role, “likely” is too much of a claim

Changed as suggested; L. 384.

- Line 424: What RPM for shaking?

We added this information (180 RPM).

- Line 486: What version of Alphafold did you use?

We added this information (AlphaFold 2).

- Line 528: How many times was it centrifuged?

We rephrased this sentence to be more precise; L. 622.

- Line 532: What were the sonication conditions?

We added this information; L. 626.

- Line 548: Is this 99.85% H₂O?

Yes, it is. We added this information; L. 642.

Reviewer #4 (Remarks to the Author):

The manuscript by Ren et al. describes the bacterial catabolism of ethylene glycol, a small alcohol derived from PET. The authors used a combination of approaches aimed at identifying EtgB and EtgA as enzymes responsible for the NAD⁺-dependent dehydrogenation of ethylene glycol and glycolaldehyde, respectively, in *Paracoccus denitrificans*. The authors further present structures of the two enzymes and investigate the transcriptional regulation of the *etg* genes. The authors also investigated the phylogenetic distribution of the pathway and used adaptive laboratory evolution to improve the strain's growth on ethylene glycol. The manuscript is clearly written and the data are clearly presented. Unfortunately, important experiments and controls are missing. Moreover, some of the data are not properly interpreted. Most importantly, the presented data do not support the conclusion that EtgB is physiologically relevant to ethylene glycol catabolism. For example, the presented data do not rule out that only one of EtgA or EtgB is involved in ethylene glycol catabolism, nor do they rule out that the PQQ-dependent methanol dehydrogenase is involved. These possibilities need to be investigated in light of the very low specificity of EtgB for ethylene glycol, the well-established role of PQQ-dependent enzymes in ethylene glycol dehydrogenation, and the known ability of glycolaldehyde dehydrogenases in improving growth of strains on ethylene glycol.

Thank you for the generally positive assessment of our manuscript. We agree that a further investigation of the roles of EtgA, EtgB, and MxaF during growth on ethylene glycol is important and have therefore performed additional experiments. In the revised version of the manuscript, we show that

- 1) The PQQ-dependent methanol dehydrogenase MxaF is not involved in ethylene glycol metabolism; *mxoF* deletion strains grow WT-like on ethylene glycol.
- 2) Both EtgA and EtgB are required for WT-like growth on ethylene glycol; *etgA* and *etgB* deletion strains exhibit strongly impaired growth on ethylene glycol.

Specific comments:

1. In the Abstract, “unknown” should be “uncharacterized” or “unrecognized” (l. 21).

We corrected this as suggested.

2. In discussing the potential of biocatalysts for upcycling plastics, the author should cite Sullivan et al. (2022; 10.1126/science.abo4626) (l. 47). This study provides proof-of-concept for the upcycling of mixed plastics waste, including PET.

We have now included this relevant reference in our introduction (L. 47).

3. The values and errors in Table 1 should be cited to the correct number of significant figures. Some values are cited to five significant figures.

We have revised Table 1 and in many cases reduced the number of significant figures.

4. Errors should be provided for the k_{cat}/K_m values (Table 1). These errors should be calculated directly from the data, not from the errors of the k_{cat} and K_m values.

We have revised Table 1 as suggested.

5. The experimental conditions (buffer, pH, temperature) used in determining the steady-state kinetic parameters should be provided as a table note (Table 1).

We have added this information as a table note to Table 1.

6. The Vmax values should be deleted from Table 1 – they are redundant with the kcat values.

We have deleted the Vmax values from Table 1.

7. In comparing the ability of EtgB to act on different substrates, the authors should compare k_{cat}/K_m values, which are a measure of substrate specificity, and not Km (ll. 163-164, l. 182).

We have revised our comparison of the activity of EtgB on different substrates by including k_{cat}/K_m values (L. 186/187, L. 191).

8. It is unclear whether EtgB oxidizes ethylene glycol significantly more efficiently than other alcohol dehydrogenases because the steady-state parameters that are compared were determined under such different conditions (p. 11). Notably, those of EtgB were determined at pH 10, its optimal pH, and not at a physiologically-relevant pH value. In this respect, the substrate specificity of glycerol dehydrogenase of *Thermus thermophilus* for ethylene glycol appears to be quite similar to that of EtgB (Table S1).

We have performed additional experiments to determine the kinetic parameters of EtgB at the physiologically relevant pH of 7.5 (see Table 1, Supplementary Table 1, and Supplementary Figure 4). While these parameters at pH 7.5 are decreased when compared to the parameters at pH 10, as expected, they are still among the best parameters determined for the NAD-dependent oxidation of ethylene glycol and allow relatively fast growth of *P. denitrificans* on this diol; the PQQ-dependent methanol dehydrogenase MxaF is not involved in ethylene glycol assimilation, as shown in the revised version of our manuscript.

The kinetic parameters of the glycerol dehydrogenase of *Thermus thermophilus* for ethylene glycol were determined at its optimal pH of 8.8 and at 70 °C, a much higher temperature than the temperature used for enzyme assays in our study (30 °C). At these optimal conditions for the glycerol dehydrogenase, its k_{cat}/K_m value of 328 is lower than the k_{cat}/K_m value of EtgB at its optimal conditions (651). The kinetic parameters of the glycerol dehydrogenase of *Thermus thermophilus* at lower temperatures have not been reported so far, but it can be expected that k_{cat} , and therefore also k_{cat}/K_m , are considerably lower at 30 °C for this enzyme. Therefore, we consider it justified to state that EtgB is among the best enzymes for the conversion of ethylene glycol to glycolaldehyde.

9. Supplementary Figure S2 needs to be cited in the text (l. 171).

We have referenced this figure (now Supplementary Figure 4) in the main text (L. 183).

10. In discussing the structural similarity of EtgB and EtgA with other enzymes, the authors should cite RMSD values or some other objective measure (p. 12). It should also be clarified which enzymes are most structurally similar to EtgB and EtgA.

We have added Supplementary Table 3 that reports the structural similarity of EtgA, EtgB, and EtgC to other proteins, as measured by RMSD.

11. Characterization of gene deletion mutants must include the phenotypes of complemented strains as a control.

Using the pREDSIX system, we generated two strains for each gene deletion in this study, with the kanamycin resistance cassette integrated in either the same or the opposite direction of transcription. By routinely generating and characterizing both strains, we made sure to exclude polar effects of gene deletions.

In the past, we have experienced decreased growth of *P. denitrificans* strains carrying replicative plasmids on some occasions (due to metabolic burden caused by the plasmid). Therefore, we chose not to generate complemented gene deletion strains, since their phenotypes cannot necessarily be expected to be identical to the WT strain, due to the observation described above.

12. When reporting Results, past tense should be used (l. 258).

The sentence in question was removed following a request from Reviewer 3.

13. The role of EtgR cannot be deduced from the growth data of the mutant alone (l. 249). The “conclusion” needs to be revised.

We agree with the reviewer that the role of EtgR cannot be deduced from the growth data of the mutant alone. Therefore, we have characterized promoter reporter strains as well, which now also includes the $\Delta etgR$ strains carrying promoter probe vectors (see Supplementary Figure 10). We evaluated growth of the $\Delta etgR$ gene deletion strains with the promoter probe vector on minimal medium with 10 mM succinate and 60 mM ethylene glycol. Their limited growth suggests that only succinate was used for growth, and the low background fluorescence as well as the lack of additional fluorescence in the presence of ethylene glycol further confirm the characterization of EtgR as an activator. If EtgR would act as a repressor, then increased background fluorescence would be expected for the $\Delta etgR$ strains with the negative control vector pTE714, and additional growth and fluorescence in the presence of succinate and ethylene glycol would be expected for all tested strains.

14. Given the high substrate specificities of EtgB and EtgA for acetaldehyde, the growth of the $\Delta etgR$ mutant strains and their complements should be tested on this substrate (Fig. 6a).

Before trying to perform the requested additional experiments, we carried out preliminary tests, aiming to grow *P. denitrificans* WT in shake flasks with different concentrations of acetaldehyde as sole source of carbon and energy. The results of these tests suggest a toxic effect of acetaldehyde on *P. denitrificans*. While barely any growth is observed at lower acetaldehyde concentrations (see below, panels a and b), some growth is observed at higher concentrations (panels c and d); however, the observed maximum OD₆₀₀ values are much lower than for acetate (panel e). Therefore, we concluded that *P. denitrificans* growth experiments on acetaldehyde cannot be carried out reliably.

15. Similarly, the promoter reporter strain should be characterized using acetaldehyde (Fig. 6b).

See above; these experiments cannot be performed reliably due to the very limited growth and observed adverse effect of acetaldehyde on *P. denitrificans*.

16. “Variants of the etg gene cluster are...” should be “The etg genes are...” (l. 314).

We revised this sentence and extended our bioinformatic analysis (L. 371-385).

17. References establishing ethylene glycol as a next-generation feedstock in microbial biotechnology should be cited in the Introduction (l. 334).

In the revised version of the manuscript, these references are now cited in the introduction (L. 51/52).

18. The statement that EtgB catalyzes the conversion of ethylene glycol to glycolaldehyde with high efficiency is incorrect (l. 341). Most importantly, the substrate specificity (k_{cat}/K_m) of EtgB for ethylene glycol is very low ($650 \text{ M}^{-1} \text{ s}^{-1}$) and the values for ethanol and acetaldehyde are ~ 300 -fold and $\sim 6,000$ -fold higher, respectively.

We have rephrased this sentence to clarify that conversion of ethylene glycol is efficient only when compared to other alcohol dehydrogenases (but not efficient when compared to the conversion of ethanol or acetaldehyde).

19. The presented data do not support the conclusion that “increased activity of EtgB and EtgA result in faster growth of the evolved strains on ethylene glycol” (ll. 344-345). For example, the inability of the ΔetgR mutant strains to grow on ethylene glycol does not test the hypothesis that only one of *etgA* or *etgB* is involved in ethylene glycol catabolism. Similarly, the analysis of the adaptive laboratory evolution strains does not test this hypothesis. Based on what is known about ethylene glycol catabolism in *Pseudomonas* and *Rhodococcus*, it is reasonable to hypothesize that in *P. denitrificans*, the PQQ-dependent methanol dehydrogenase and EtgA catalyze the dehydrogenation of ethylene glycol and glycolaldehyde, respectively. Glycolaldehyde is cytotoxic, and its transformation is rate-limiting in many strains. This hypothesis is consistent with the adaptive laboratory evolution data in *P. denitrificans* (i.e., increased production of EtgA would decrease the intracellular concentration of glycolaldehyde).

To address the hypothesis that the PQQ-dependent methanol dehydrogenase and EtgA catalyze the dehydrogenation of ethylene glycol and glycolaldehyde, respectively, we have generated gene deletion strains of *etgA*, *etgB*, and *mxoF*. Our results (see Supplementary Figures 2 and 3) demonstrate that 1) *MxoF* is not involved in ethylene glycol assimilation and 2) both EtgA and EtgB are required for WT-like growth on ethylene glycol. Therefore, the hypothesis that EtgA and EtgB are the key enzymes for oxidation of ethylene glycol to glycolate is further supported.

Thank you for submitting your manuscript "NAD-dependent dehydrogenases enable efficient growth of the soil bacterium *Paracoccus denitrificans* on the PET monomer ethylene glycol" to Nature Communications. I am delighted to say that we are happy, in principle, to publish it under an open access license.

First, we ask you to revise your paper to address our editorial requests (in the attached Author Checklist) and any remaining comments from reviewers.

We are very glad to hear that our revised manuscript is close to acceptance in Nature Communications now, and we are grateful to the Reviewers 1, 2, and 4 for their positive feedback. We have addressed the additional comments from Reviewer 3 in detail below, and we have made changes according to the editorial requests, which are documented in detail in the Author Checklist (uploaded as separate document).

Reviewer #1:

My concerns have been addressed. Thanks authors' efforts.

Reviewer #2:

The authors have nicely revised their manuscript and addressed all the concerns that I had raised in the previous round of review.

Reviewer #3:

I am pleased to see the authors have taken the time to address all the comments in our original rebuttal. The study and manuscript are now much more complete, coherent and impactful. I therefore recommend publication after the authors address the following minor remaining edits:

- Error bars should be included in all growth curves. This can be made less distracting by using a light coloration relative to the plotted data and will be more informative to illustrate any significant differences to the reader.

We have modified all figures showing growth curves as suggested.

- the authors mention that the majority of K_m values in Table 1 are in the low μM range, but 11 of the 20 are in the mM range. I think this should be changed to mM for consistency with Supplementary Table 1.

We have changed Table 1 as suggested.

- Line 52: as a source of carbon

Corrected as suggested.

- Line 101: which then enters central carbon metabolism...

Corrected as suggested.

- Line 186: with a k_{cat}/K_m value that is 300-fold lower.

Corrected as suggested.

Reviewer #4:

The revised manuscript contains a considerable amount of new data and analyses that address my comments on the original submission. This more complete study significantly advances our understanding of the bacterial catabolism of ethylene glycol.